# Structural basis for the allosteric modulation of rhodopsin by nanobody binding to its extracellular domain

Arum Wu [1,12], David Salom[1,12], John D. Hong[1,2], Aleksander Tworak [1], Kohei Watanabe[3,4], Els Pardon [5,6], Jan Steyaert [5,6], Hideki Kandori [3,7], Kota Katayama [3,4,7] ✉, Philip D. Kiser [1,8,9,10] ✉ & Krzysztof Palczewski [1,2,8,11] ✉

Rhodopsin is a prototypical G protein-coupled receptor (GPCR) critical for vertebrate vision. Research on GPCR signaling states has been facilitated using llama-derived nanobodies (Nbs), some of which bind to the intracellular surface to allosterically modulate the receptor. Extracellularly binding allosteric nanobodies have also been investigated, but the structural basis for their activity has not been resolved to date. Here, we report a library of Nbs that bind to the extracellular surface of rhodopsin and allosterically modulate the thermodynamics of its activation process. Crystal structures of Nb2 in complex with native rhodopsin reveal a mechanism of allosteric modulation involving extracellular loop 2 and native glycans. Nb2 binding suppresses Schiff base deprotonation and hydrolysis and prevents intracellular outward movement of helices five and six – a universal activation event for GPCRs. Nb2 also mitigates protein misfolding in a disease-associated mutant rhodopsin. Our data show the power of nanobodies to modulate the photoactivation of rhodopsin and potentially serve as therapeutic agents for disease-associated rhodopsin misfolding.

Rhodopsin (Rho), composed of seven transmembrane domains, a characteristic of all G protein-coupled receptors (GPCRs), and its covalently bound chromophore 11-*cis*-retinylidene[1], is a major component of the intracellular disk membranes of rod photoreceptor cells[2]. Upon photoisomerization of its chromophore from 11-*cis*- to all-*trans*-retinylidene, the photoactivated rhodopsin (Rho*) undergoes a complex sequence of structural changes, leading to the signaling state of the receptor, metarhodopsin II (Meta II)[3]. In this prototypical,

precisely timed, and mostly irreversible process, the ultrafast photo-isomerization of rhodopsin is followed by transitions through various short-lived intermediates leading to the formation of the fully activated Meta II species before the Schiff-base-adducted chromophore is hydrolytically released[4,5]. These events can be monitored by UV-Vis absorption spectroscopy because the bound chromophore is spectrally sensitive to its changing environment as the opsin progresses through its different conformers. Upon activation of the GPCR,

[1]Department of Ophthalmology, Gavin Herbert Eye Institute, University of California, Irvine, CA 92697, USA. [2]Department of Chemistry, University of California, Irvine, CA 92697, USA. [3]Department of Life Science and Applied Chemistry, Nagoya Institute of Technology, Showa-ku, Nagoya 466- 8555, Japan. [4]PRESTO, Japan Science and Technology Agency, 4-1-8 Honcho, Kawaguchi, Saitama 332-0012, Japan. [5]Structural Biology Brussels, Vrije Universiteit Brussel, Brussels, Belgium. [6]VIB-VUB Center for Structural Biology, VIB, Brussels, Belgium. [7]OptoBioTechnology Research Center, Nagoya Institute of Technology, Showa-ku, Nagoya 466-8555, Japan. [8]Department of Physiology & Biophysics, University of California, Irvine, CA, USA. [9]Department of Clinical Pharmacy Practice, University of California, Irvine, CA, USA. [10]Research Service, VA Long Beach Healthcare System, Long Beach, CA, USA. [11]Department of Molecular Biology and Biochemistry, University of California, Irvine, CA 92697, USA. [12]These authors contributed equally: Arum Wu, David Salom. ✉e-mail: katayama.kota@nitech.ac.jp; pkiser@uci.edu; kpalczew@uci.edu

different conformations can also be observed by Fourier-transform infrared (FTIR) spectroscopy[6,7]. These spectral properties of Rho distinguish it from other GPCRs, which respond to diffusible chemical ligands instead of light. The light-triggered spectral transitions of the Rho activation cascade provide a prototypical road map for how GPCRs adopt the ligand-induced active conformations capable of activating G proteins.

Since the first high-resolution structure of Rho was reported in 2000[1], advances in the structural biology of GPCRs have been remarkable. Structural characterization of various forms of the receptors, including native or mutated forms, engineered fusion constructs with other proteins, and complexes with interacting partners or antibody fragments, have provided critical insights into the chemistry and pharmacology of these receptors[8–12]. The activation process of Rho is one of the best understood among the GPCRs[13–15]. Structures of several intermediates in the photoactivation process have been solved by crystallography or by cryo-electron microscopy (cryo-EM). However, neither X-ray crystallography nor cryo-EM can unambiguously assign the observed structures to particular signaling states, unless we assume that the structures captured in crystals or frozen on grids faithfully reflect corresponding conformations found in solution or in native membranes.

Mutagenesis has been used extensively to infer structural details of the Rho activation process, although this approach has limitations. For example, it is well established that even a single point mutation in the Rho sequence can significantly affect the dynamics of Rho* in its transition through intermediates[16]. In most cases, the mutations change the opsin-chromophore interactions, potentially causing rearrangement of the hydrogen-bond network of water molecules within the transmembrane segments of the protein[17,18], a critical element of the unique thermodynamic state of Rho. A different approach to gain structural insights involves the use of external ligands including proteins to stabilize specific forms of Rho*. This approach does not perturb the intrinsic energetics of Rho* formation, as it typically would leave the exquisitely sensitive water network formed in the transmembrane domain unmodified. Instead, the added proteins stabilize the overall protein conformation, thereby capturing a specific state that can be studied structurally and biophysically. In this work, aiming to better understand the conformational landscape of Rho following photoactivation, we develop and characterize a series of camelid nanobodies (Nbs) that can modulate the conformation of Rho* through binding to its extracellular loop II (EL2) and the N-terminus, including the two glycans at Asn[2] and Asn[15]. Using these reagents, we carry out biochemical, structural, spectroscopic, and cellular studies to identify a region of the Rho structure that plays a pivotal role in the process of switching between different conformational states.

## Results

### Screening bRho-specific binding of Nbs to both inactive and active-state Rho

Nbs against bovine Rho (bRho) were raised by injections of purified ROS (rod outer segment) into a llama. Twenty-seven Nb clones targeting native bRho were identified by ELISA. To verify the binding of the Nbs to bRho, we performed small-scale affinity chromatography with His-tagged Nbs, immobilized onto Ni[2+]-NTA resin. Nineteen Nbs were identified to bind Rho in the light (Supplementary Fig. 1a, upper) or the dark (Supplementary Fig. 1a, lower), suggesting recognition of both activated and ground-state conformations of bRho. Further screening for high-affinity Nbs was done by blue-native PAGE (BN-PAGE) in the dark, a more stringent assay for protein-protein interactions compared to affinity chromatography. Among the 19 Nbs, four (Nb2, Nb7, Nb12, and Nb22) produced gel shifts of bRho in a dose-dependent manner (Supplementary Fig. 1b). Based on the amount of Nb needed for complete gel shift of bRho, the relative strengths of Nb binding to bRho exhibited the following order: Nb7 > Nb12 > Nb2 =

Nb22. The electrophoretic mobilities of the bRho/Nb complexes relative to bRho alone were found to correlate with the predicted isoelectric points (pI) of the Nbs; the pIs of Nb2, Nb7, Nb12, and Nb22 are 6.64, 7.18, 7.18, and 8.01, respectively. Nb22 produced an additional super-shift band in a dose-dependent manner, likely due to the formation of dimers or aggregates (Supplementary Fig. 1b, bottom panel). Next, we performed BN-PAGE analysis of the bRho/Nb complexes under both dark and light (Supplementary Fig. 1c) conditions to test for Nb-affinity differences for ground-state *versus* bRho*. Under illuminated conditions, co-migration of bRho with Nbs was not detected, indicative of a stronger binding of the Nbs to ground-state bRho. A neighbor-joining tree of the Nb sequences showed Nbs 2, 7, and 12 clustering on one side of the central internal branch, with Nbs 7 and 12 differing by a single amino acid in CDR1 and having ~78% sequence identity to Nb2 (Supplementary Fig. 1d, e). Poorly binding or nonbinding Nbs including Nb16 were found on the opposite side of the tree with sequence identities of 63–71 % to Nb2. Nb9 and Nb17, moderate binders of bRho, clustered closely with Nb22 and diverged from the good binders, supporting the distinctive behavior of Nb22 on BN-PAGE.

### The crystal structure of ground-state Rho in complex with Nb2 reveals a conformational epitope composed of the N-terminus and extracellular loop II

Among the 4 Nbs with high affinity for bRho, we were able to determine the crystal structure of ground-state bRho in complex with Nb2 at a resolution of 3.7 Å (Supplementary Table 1). The asymmetric unit of these crystals contained a parallel bRho dimer, like previously observed bRho dimers[19–21], with Nb2 bound to both protomers at their extracellular surfaces (Fig. 1a, Supplementary Fig. 2). Inspection of the bRho/Nb2 interface revealed close contacts of CDR1 and CDR3 of Nb2 with the C-terminal region of EL2 of bRho (Fig. 1b, c). We observed that Phe[27], Thr[28], Lys[31], and Tyr[32] in CDR1 of Nb2 interact with Pro[194], His[195], Glu[196], and Glu[197] of EL2 in bRho via electrostatic and van der Waals interactions. Glu[197] of EL2 and Glu[201] located at the top of TM5 also formed electrostatic interactions with Arg[98] of CDR3 and Lys[31] of CDR1 of Nb2. In addition to these interactions with EL2 and TM5 of bRho, Gly[99], Tyr[100], Gly[101], and Met[103] of Nb2-CDR3 and Asp[62] and Trp[47], which are outside the classical CDR regions of Nb2, interact with the N-terminus of bRho (Fig. 1c–f). Tyr100 of Nb2 also interacts with Asn279 and Gly280 in EL3 of bRho.

Alanine scanning mutagenesis of CDR1 and CDR3 revealed the residues most critical for antigen recognition. Among the 17 point-mutations screened, six (F27A, Y32A, R98A, G99A, Y100A and G101A) decreased the Nb2 binding affinity to bRho more than 80 % compared to WT Nb2 (Fig. 1d, Supplementary Fig 3a). Our screening results were mapped onto a space-filling model of Nb2, depicting that the six most critical amino acid residues in CDR1 and CDR3 are tightly intertwined, constituting a single conformational paratope (Fig. 1e, highlighted in red). As shown in Fig. 1f, the placement of these six critical residues of Nb2 near the extracellular surface of bRho revealed a binding "hot spot" on EL2 consisting of Pro[194], His[195], Glu[196], and Glu[197].

To probe this putative EL2 hotspot, we performed a co-immunoprecipitation (co-IP) assay to investigate the binding of Nb2 to mouse Rho (mRho), which contains non-conservative amino acid substitutions at positions 194-196, and 198 (Supplementary Fig. 3b). We found that Nb2 (Supplementary Fig. 3c), as well as Nbs 7, 12, and 22 (data not shown) failed to bind wild-type (WT) mRho, further supporting the critical antigenic nature of the EL2 hotspot of bRho. Substitution of the WT mRho residues at these 4 positions with those of bRho (m/bEL2 hybrid) restored the antigen-antibody interaction for Nbs 2, 7, 12, and 22. These results confirmed the crucial nature of EL2 in bRho in determining high affinity binding with this series of Nbs (Fig. 1g).

The crystal structure of the bRho/Nb2 complex shows that CDR2, CDR3, and the loop connecting the C and C' strands (FR2) of Nb2

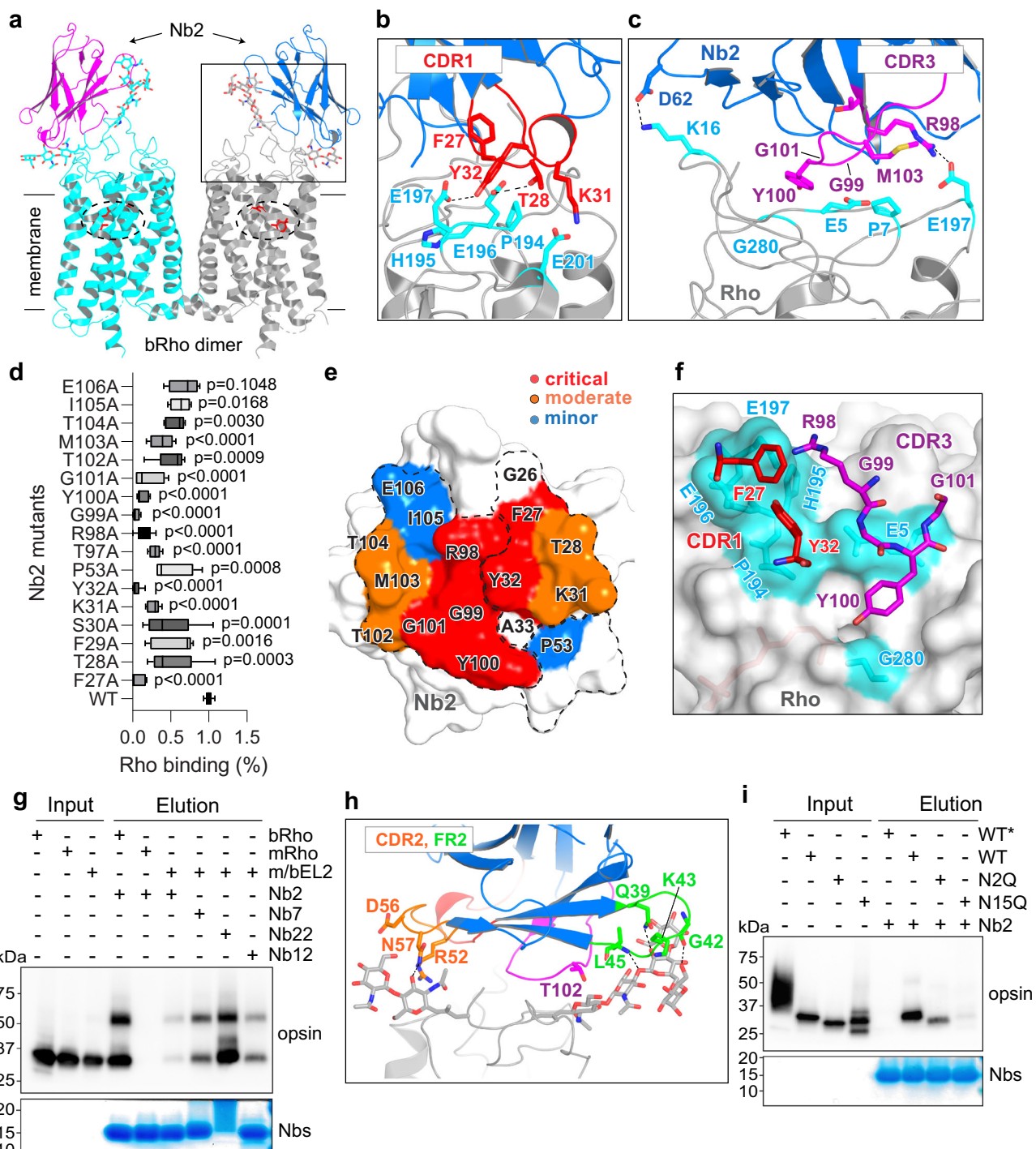

**Fig. 1 | Crystal structure of ground-state bRho in complex with Nb2, and validation of the epitope- paratope interaction surfaces. a** Parallel arrangement of bRho/Nb2 heterotetramer found in the asymmetric unit. The retinal binding pocket is encircled with 11-*cis*-retinal, shown in red sticks. In **b**, **c**, **f** and **h**, the bRho and Nb2 binding interface is boxed and analyzed in detail. The CDR1, CDR2, CDR3, and FR2 regions of Nb2 are labeled in red, orange, magenta, and green, respectively, and corresponding contact residues in Rho are shown in cyan. **b**, **c** Side views of the bRho-Nb2 interaction surface. Glycans are omitted for clarity. **d**–**f** Mapping of hotspots between bRho and Nb2 by alanine scanning mutagenesis of residues in the CDRs of Nb2. The binding of the resultant mutants to bROS was assayed by co-immunoprecipitation with bROS, followed by western blotting with anti-Rho 1D4 antibody (**d**). Data have been normalized and presented as the percentage of WT Nb2 values for Rho binding. Data are presented as mean values +/- SD from n = 4 or 5 biologically independent experiments. The center line represents the median value,

and the boundaries indicate the 25th percentile and the 75th percentile. Whisker boundaries indicate the data range. One-way ANOVA was used to calculate the statistical significance of the differences in Rho binding for WT *versus* Nb2 mutants. **e** The alanine-scanning mutagenesis results from (**d**) were mapped onto the three-dimensional, space-filled structure of Nb2. **f** Extracellular view of bRho with superimposed critical residues of Nb2. **g** Co-IP of Nbs with recombinant bRho, mRho, and m/bEL2 (mRho with bovine EL2). Equal amounts of Rho samples (Input) were pulled down with His-tagged Nbs (IMAC) and eluted with an excess amount of imidazole. Opsins were viewed by Western blotting with anti-Rho 1D4 antibody and Nbs were detected by Coomassie blue staining. Uncropped blots are provided in the Source Data. **h** Side view of the bRho-Nb2 interaction surface *via* glycans. **i** Co-IP of Nb2 with recombinant bRho, overexpressed in hyper-glycosylating NIH3T3 cells (bWT*), GnTI- HEK293S cells (bWT) or N2Q- and N15Q-mutant bRho-overexpressing HEK293S cells. Source data for panels **d**, **g**, and **i** are provided as a Source Data file.

interact with the two N-linked glycans of bRho via hydrogen bonding and van der Waals interactions with Gly[42], Lys[43], Gly[44], Leu[45], and Thr[102] (Asn[15]-linked glycan) and Arg[52], Asp[56], and Asn[57] (Asn[2]- linked glycan) (Fig. 1h). Indeed, the Nb2 surface area buried by these glycans (~512 Å$^2$) is of the same order as that buried by amino acid residues (~612 Å$^2$), suggesting that the interactions may significantly contribute to the strength of the bRho-Nb2 interaction. To test this hypothesis biochemically, we substituted Gln for Asn either at positions 2 or 15 in bRho to eliminate glycosylation at these sites and expressed each mutant opsin in HEK293S GnTI⁻ cells[22]. Previous studies showed that the removal of N-linked glycosylation affects the stability and folding of bRho[23]. We observed a similar lower-level expression of N15Q-opsin necessitating five times the number of cells to yield an equivalent amount of protein as that for WT. Using a co-IP assay, we observed that Nb2 had little or no interaction with N2Q-opsin and an even weaker interaction with N15Q-opsin, consistent with our observation that the Asn[15]-linked glycan has the greatest interaction with Nb2. As an additional test, we expressed WT-bRho in NIH3T3 cells, which are known to hyperglycosylate Rho, and found that hyperglycosylation also abolished interaction of Nb2 with bWT-opsin (Fig. 1i). These findings corroborate our structure-based prediction that N- linked glycans serve as critical epitopes, contributing to the specificity of Nb2 antigen recognition (Fig. 1i, lane 7 and 8). Collectively, the crystal structure of the bRho/Nb2 complex together with our validating mutagenesis and biochemical studies confirm the discontinuous conformational epitope of bRho, which consists primarily of EL2 and N-terminal glycans.

To further quantify the affinity of the bRho/Nb2 complex, we performed surface plasmon resonance (SPR) with varied concentrations of purified bRho passed over Nb2 that was immobilized on the sensor surface. Analysis of the SPR sensogram revealed that Nb2 binds to bRho with nanomolar affinity ($K_D = 117 \pm 1.2$ nM), with an on-rate constant ($k_{on}$) of $4.01 \times 10^4$ M$^{-1}$ s$^{-1}$ and an off-rate constant ($k_{off}$) of $4.67 \times 10^{-3}$ s$^{-1}$ (Fig. 2a). Fitting of the data to a 1:1 Langmuir model yielded a similar $K_D$ of 148.7 nM (Fig. 2b).

## Nb binding stabilizes Meta I-like intermediate conformations in detergent solution

Next, we employed UV-Vis spectroscopy to analyze how environmental factors such as temperature and pH affect the stability of bRho*/Nb2 complexes. We measured bRho decay after illumination of the bRho/Nb2 complex in detergent solution (Fig. 2c). Following illumination, the resultant bRho* undergoes a series of conformational changes to spectrally distinct intermediates[24] (Fig. 2d). Of these, Meta I (λmax ~ 478 nm) and Meta II (λmax ~ 380 nm) are relatively long-lived and exist in an equilibrium that favors Meta II at physiological pH. After forming the bRho/Nb2 complex in the dark, the transformation and decay of bRho*/Nb2 complexes were measured at different temperatures (Fig. 2e, f) and at various pH values (Fig. 2g, h). The same measurements were taken without Nb2 as controls for these experiments (Supplementary Fig. 4). Surprisingly, Nb2 was observed to shift the equilibrium of the photointermediates towards the Meta I-like species (Fig. 2e), thereby slowing the overall decay of bRho* to apo-opsin in a temperature-dependent manner (Fig. 2f vs. Supplementary Fig. 4a). Constant illumination of the bRho*/Nb2 complex further enhanced decay of the optical signal at 478 nm, likely due to enhanced formation of meta II and consequent loss of retinal chromophore and/or its photolysis (Fig. 2f). The stability of bRho*/Nb2 also demonstrated pH dependency with greater stability over time at higher pH, favoring the Meta I-like species (Fig. 2g, h vs. Supplementary Fig. 4b). bRho* alone in dark conditions showed a small increase in absorbance at 478 nm over time, possibly arising from stable Meta I or Meta III-like photoproducts (Supplementary Fig. 4a, b, day 11). Collectively, these differences in UV-Vis spectral properties of bRho with and without Nb2 indicate that Nb2 binding stabilizes bRho* in a Meta I-like state upon photoactivation in a detergent solution.

## Identification of Lumi-like states of Rho/Nb2 complex in membranes by light-induced ATR-FTIR spectroscopy

To gain further insights into the identity of the bRho photointermediate(s) stabilized by Nb2, we subjected the bRho/Nb2 complex in native membranes to Attenuated Total Reflectance - Fourier Transform InfraRed (ATR- FTIR) difference spectroscopy, comparing before *versus* after a light exposure (Fig. 3a). This technique provides structurally-sensitive, feature-rich difference spectra that can distinguish among the various photoactivated states of Rho based on functional group-specific vibrational absorption bands[7,25]. It is noted that the spectra of the Lumi and Meta I intermediates have been obtained previously by cryo-trapping at 200 K and 240 K, respectively, while the measurement of the complex with Nbs was performed at room temperature using an ATR-FTIR spectrometer. FTIR difference spectra of bRho/Nb2 and bRho/Nb9 (Fig. 3b, red curves) resembled the Lumi and Meta I reference spectra measured at 200 K and 240 K, respectively (Fig. 3b, blue and grey curves). The Lumi and Meta I difference spectra of the bRho*/Nb complexes were closely related due to the reduced intensities of positive bands upon Nb binding at 240 K, but they can be distinguished by differences in the fingerprint bands in the range of 1800 and 1600 cm$^{-1}$ (Fig. 3b, highlighted in yellow). Vibrations from the chromophore appear in the 1300−900 cm$^{-1}$ region. The characteristic positive hydrogen-out-of-plane (HOOP) bands for Lumi and Meta-1 appear at 947 (+) cm$^{-1}$ and 951 (+) cm$^{-1}$, respectively, but not in Meta II Rho due to deprotonation of the Schiff base. The negative bands at 967 (−) cm$^{-1}$ and 1238 (−) cm$^{-1}$ in Nb2 and Nb9-bound bRho* originate from a HC11 = C12H HOOP mode and C-C stretching, respectively, in native Rho while positive 1206 cm$^{-1}$, 1198 cm$^{-1}$, and 1184 cm$^{-1}$ bands are seen for Lumi and Meta I, but not Meta II[26–31]. Prior reports showed that 1656 (−) bands (C = O stretch, Amide-I) appear in both Lumi and Meta I but not in Batho, suggesting that bRho*/Nb2 and bRho/Nb9 complexes are in the Lumi or Meta I states but not in the Batho state. According to the reports, the negative 1664 (−) band in Batho gradually increases into a baseline in the Lumi state and into a characteristic 1664 (+) fingerprint band during the transition to Meta I. The 1664 (+) Meta- I fingerprint band was not observed in the spectrum of bRho*/Nb2 but was present in that of bRho*/Nb9. The band at 1636 (+) cm$^{-1}$ (C = O stretch, Amide I) in the spectra of bRho*-Nb2 is a Lumi-specific band, which exists as a baseline 1636 (+)/1643 (+) cm$^{-1}$ doublet in bRho*-Nb9 and Meta-I reference spectra. The bRho*/Nb2 complex showed bands at 947 (+) cm$^{-1}$ as well as a paired band at 1656 (−) /1636 (+) cm$^{-1}$, fitting better with the reference difference spectrum of Lumi than that of Meta I or II. The Meta II bands at 1656 (−)/1644 (+) cm$^{-1}$, 1713 (+) cm$^{-1}$ and 1769 (−)/1747 (+)/1728 (−) cm$^{-1}$ were not observed in the spectrum of the Nb2-bound Rho. The positive 1713 cm$^{-1}$ band, which arises from protonation of both Glu[113] and Glu[134] as the ionic lock[32,33] is disrupted, was not observed in the bRho*/Nb2 or bRho*/Nb9 difference spectra. When we compare the difference FTIR spectra of bRho* in complex with other Nbs, bRho*/Nb17 most closely resembled Meta I, based on characteristic bands at 1664 (+) cm$^{-1}$ but absence of the 1636 (+) cm$^{-1}$ band, while a majority of Nb- bound Rho FTIR spectra were closely related to Lumi reference spectra (Supplementary Fig. 5a, b). Unsurprisingly, non-binding Nbs (Nb3, Nb10 and Nb16) showed high similarity to Meta II reference spectra, indicating that they lack the ability to modulate progression of the photocycle. These results demonstrated that subsets of Nbs could stabilize bRho* in the Lumi or Meta I photointermediate states, rather than Meta II active state.

## Nb2 shifts the equilibrium of Rho* from Meta II to Meta I-like in detergent solution

Because Nb2 recognizes both ground-state and photoactivated bRho (Supplementary Fig. 1a), we investigated by UV-Vis spectroscopy the spectral changes of bRho* that occur following the addition of Nb2 at various pHs at 0 °C (Fig. 3c). The spectrum of bRho* alone in detergent

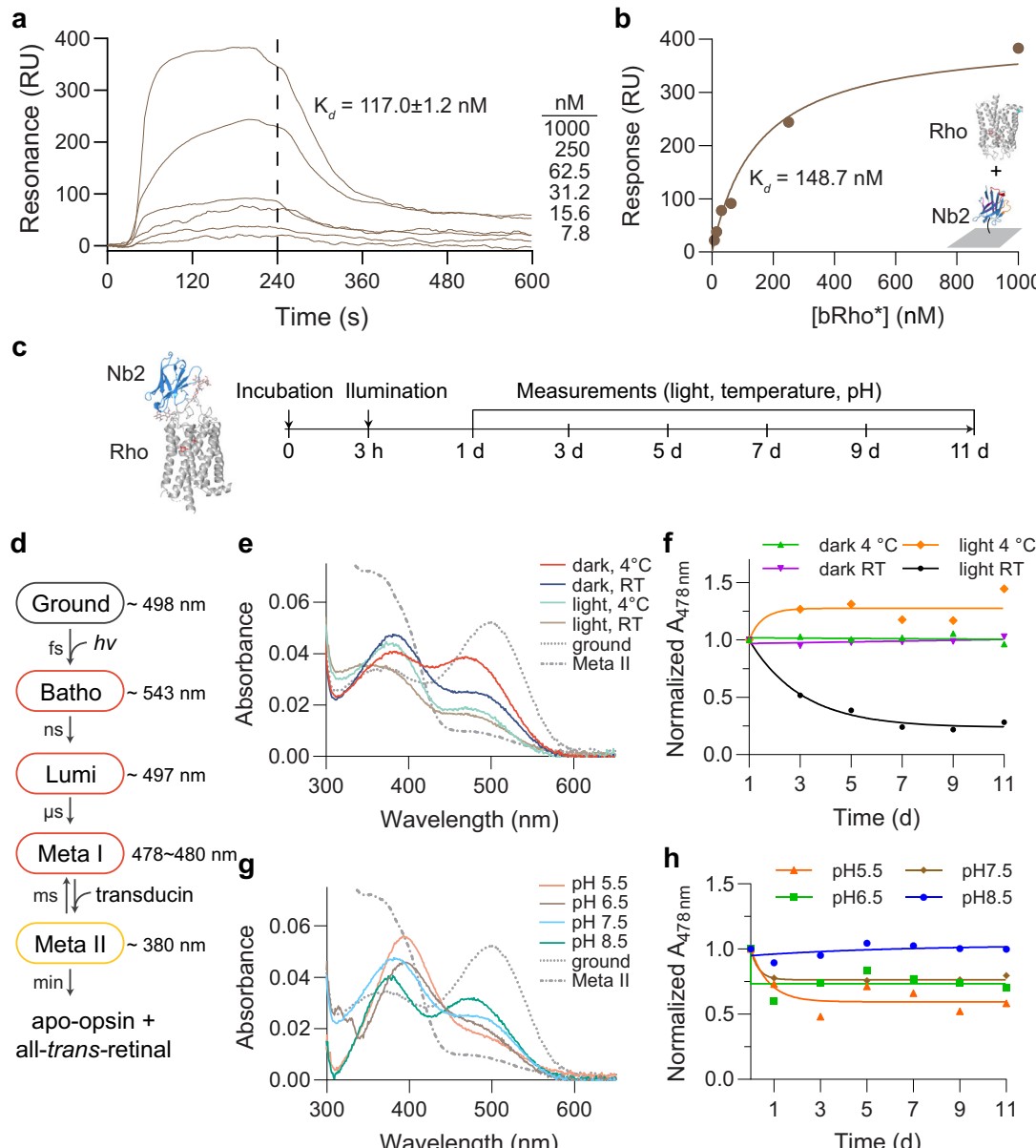

**Fig. 2 | High affinity binding of Nb2 to bRho stabilized a Meta I-like state following photoactivation. a, b** Surface plasmon resonance (SPR) binding equilibrium of bRho to immobilized Nb2 was analyzed by kinetic analysis (**a**) and affinity-based analysis (**b**). Each approach was used independently to determine the $K_d$ value. **c–g** Experimental scheme (**c**) for decay of bRho/Nb2 complexes after white-light illumination for 3 min. Ground- state bRho was incubated with Nb2 for 3 h in the dark before illumination, followed by UV-Vis spectroscopic measurements for 11 days, under light/dark and different temperatures (**e**, **f**); or different pHs in the dark at RT (**g**, **h**). Corresponding data for bRho alone are shown in Supplementary Fig. 4. **d** The signaling states of bovine Rho during photoactivation. **e** Representative UV-Vis absorbance spectra of bRho/Nb2 complexes measured at day 11.

Ground-state and Meta II bRho are shown in dotted grey lines for comparison. **f** UV-Vis spectra of Rho/Nb2 complexes in light/dark and 4 °C/RT conditions were recorded at various times for the time course. A478 nm values were normalized to that of bleached bRho/Nb2 at time zero. **g** Representative UV-Vis absorbance spectra of bRho/Nb2 complexes measured at day 11. Inactive (ground) and activated (Meta II) Rho are shown in dotted grey lines for comparison. **h** UV-Vis spectra of bRho/Nb2 complexes at various pH conditions (pH 5.5–8.5) were recorded as a function of time. A478 nm values were normalized to the value immediately after illumination. Source data for panels **a**, **b**, and **e**–**h** are provided as a Source Data file.

solution was dominated by the Meta II-associated absorbance band at 380 nm, although a small amount of Meta I signal at 478 nm was still present following a 30-sec illumination (Supplementary Fig. 5c, t = 0). The addition of Nb2 to bRho* caused a shift from Meta II to Meta I-like intermediates, resulting in an increase of A478 nm at physiological pH values (Fig. 3d, Supplementary Fig. 5c). The ability of Nb2 to shift Meta II to the Meta I-like conformation was clearly observable by the color change from pale-yellow (Meta II) to orange (Meta I-like) after addition of Nb2 to bRho* (Fig. 3e). Other Nbs including Nb7, Nb12 and Nb22 also showed similar equilibrium shifting of bRho* from Meta II to Meta I-like

states (Supplementary Fig. 5d, e). In summary, addition of Nbs to bRho* shifted the equilibrium towards Meta I-like photointermediates in detergent solution (Fig. 3f).

**Structural comparison of Rho/Nb2, Rho*/Nb2, apo-opsin/Nb2, to ground-state bRho and apo-opsin**
Prior reports showed that the Batho, Lumi, Meta I, and Meta IIa structures are very similar to that of ground-state bRho, and that EL2 plays a critical role in helix movements during bRho activation[19,34–37]. Accordingly, we hypothesized that Nb2 binding to EL2 stabilized Lumi-

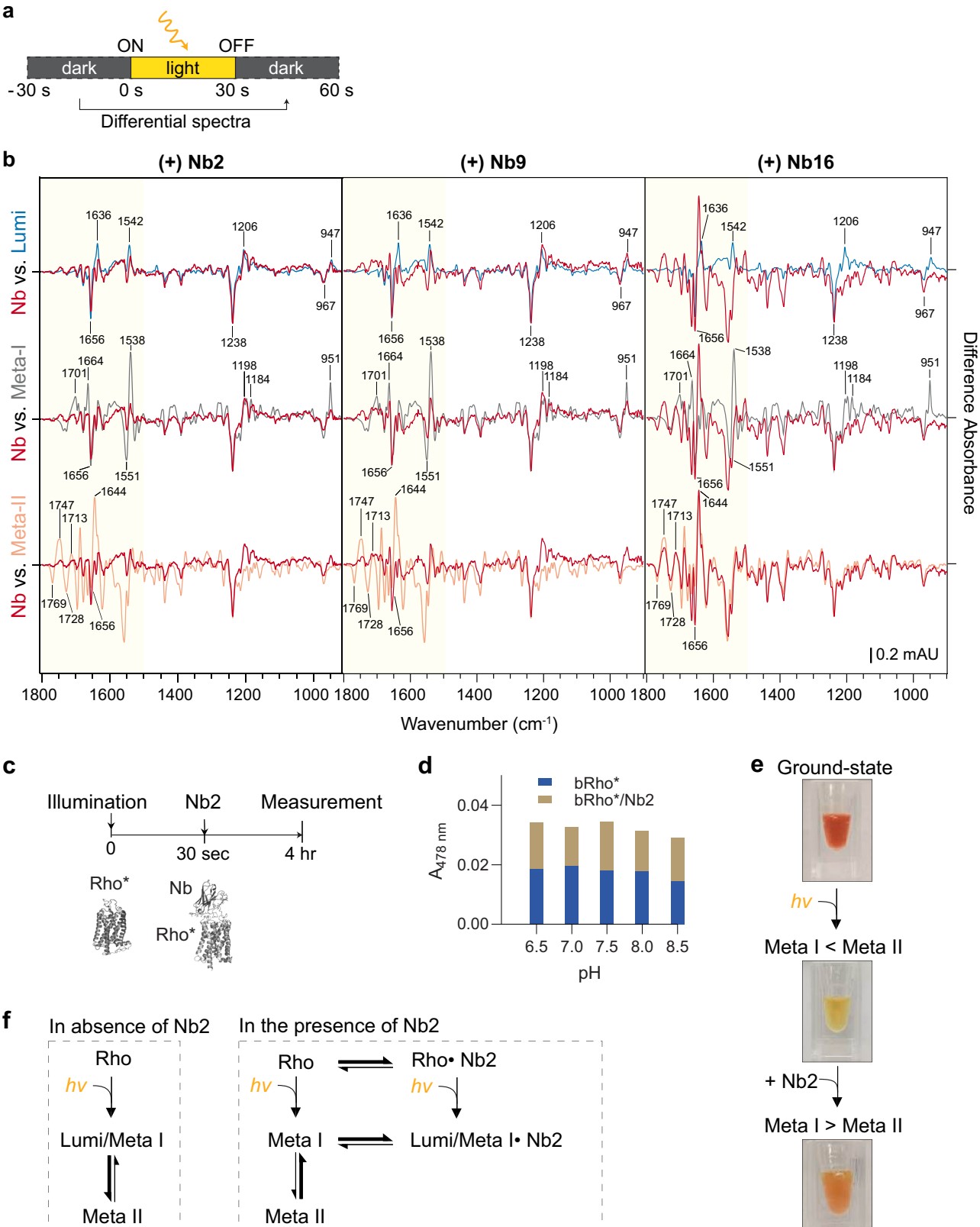

or Meta I-like intermediates upon photoactivation by locking the EL2 plug of the chromophore binding pocket, preventing large structural changes on the cytoplasmic side. We tested this hypothesis by comparing crystal structures of ground-state bRho, bRho*, and opsin in complex with Nb2 to a published structure of bovine opsin. To obtain opsin/Nb2 and bRho*/Nb2 crystals, we took two approaches. First,

crystals of ground-state bRho in a complex with Nb2 were photo-activated by green light, resulting in a change in their color from red to orange (Supplementary Fig. 6a), mimicking the experimental conditions where pre-formed bRho/Nb2 complexes were trapped as Meta I-like photoproducts after illumination. Second, to obtain opsin/Nb2 crystals, bRho/Nb2 complexes were bleached in the presence of

**Fig. 3 | Rho in complex with Nb2 confirms the stabilization of a Lumi-like conformation in membranes upon photoactivation. a** Experimental scheme for light-induced ATR-FTIR difference spectroscopy in ROS membranes. Difference spectra were calculated (*after* minus *before* illumination). **b** FTIR difference spectra of bRho* alone and bRho* with Nb2, Nb9, or Nb16 (red curves). The spectra for Lumi (blue curves), Meta I- (grey curves) and Meta II-bRho (orange curves) were recorded at 200 K, at 240 K, and at 277 K, respectively. Negative bands correspond to vibrations present in the ground-state bRho, and positive bands correspond to vibrations in photoactivated bRho. Major spectral changes are seen in highlighted regions between 1500–1800 cm$^{-1}$ (yellow). **c, d** Experimental scheme (**c**) to monitor the equilibrium shift of Rho*/Nb2 complex in detergent solution. **d** bRho* alone or in complex with Nb2 was incubated on ice for 4 h at different pHs followed by

dilution and measurement of absorbance (Supplementary Fig. 5c). A478 nm values from the corresponding UV-Vis spectra of bRho*/Nb2 (mustard yellow) and bRho* alone (blue) were plotted. **e** Visual depiction of light-activated Rho* pigments from pale-yellow (dominantly Meta II) to orange (Meta I-like) upon Nb2 binding, indicating a shift in the equilibrium. **f** Mechanisms of Nb2 stabilization of the Meta I-like conformation by binding to heterogenous Rho (inactive *vs*. intermediate). In the absence of Nb2, illumination triggers conversion of bRho from ground-state to Meta I followed by Meta II. In the presence of Nb2, Rho is trapped in Lumi/Meta I-like intermediate states upon illumination. When Nb2 is added to photoactivated bRho, the preexisting equilibrium of Meta I/Meta II, Rho* shifts towards Meta I-like intermediates. Source data for panels **b** and **d** are provided as a Source Data file.

hydroxylamine (NH$_2$OH) to induce hydrolysis of the Schiff base, followed by crystallization.

The structure of the ground-state bRho/Nb2 complex was virtually identical to that of bRho alone (Supplementary Fig. 6b, c). Superposition of bRho/Nb2 (red), bRho*/Nb2 (orange), and opsin/Nb2 (cyan) revealed overall similar arrangements (Fig. 4a). We did not detect any major movements of the binding interface between Nb2 and the extracellular loops of bRho* or opsin compared to that of bRho/Nb2, as the retinal plug formed by EL2 remained in place over the orthosteric binding pocket (Fig. 4b). Within the transmembrane bundle, the bRho/Nb2, bRho*/Nb2, and opsin/Nb2 structures do not differ on the intracellular side, and they show only a slight tilt of the cytoplasmic end of TM7 (Fig. 4c). By contrast, the comparison of the opsin/Nb2 structure (Fig. 4d, cyan) to a published apo-opsin crystal structure[38] (Fig. 4d, dark blue) showed clear differences in the extracellular loops, and outward tilting of TM5 and TM6 on the intracellular side (Fig. 4d, e, black arrows). In the opsin structure, the outward tilting of TM5 and TM6 occurs concomitantly with slight rotation of TM3 and TM6, causing widening of the binding pocket and enabling water molecules to hydrolyze the Schiff base at Lys$^{296}$, thereby breaking the ionic lock between Arg$^{135}$ on TM3 and Glu$^{247}$ on TM6, as shown in Fig. 4f (black arrows). The E(D)RY ionic lock appears to remain intact in the bRho/Nb2, bRho*/Nb2, and opsin/Nb2 structures, as no rotation of Glu$^{134}$ on TM3 or Glu$^{247}$ on TM6 was observed (Fig. 4f). This observation differs from the published opsin crystal structures, where Glu$^{247}$ on TM6 is displaced and closely juxtaposed to Lys$^{231}$ on TM5, forming a new electrostatic interaction and stabilizing TM5-TM6 in an active conformation. Collectively, our current crystallographic data indicate that Nb2 binding to ground-state bRho, bRho*, and opsin likely confers rigidity to the extracellular side of the receptor via EL2 and N-terminal (including N-linked glycans) interactions, which in turn prevent movements of the transmembrane helices, breakage of the ionic lock, and conversion of the activation switch from Meta I to Meta II upon bRho photoactivation.

## Hydrolysis of the Schiff base of Rho* is slowed by the complex formation with Nb2

Prior reports have shown that EL2 plays an important role in controlling solvent access to the retinal binding site and consequent hydrolysis of the all-*trans*-retinylidene deprotonated Schiff base[34]. Additionally, it was shown that mutations in EL2 accelerated hydrolysis and reduced the thermal stability of inactive- and active-state Rho[39–41]. Thus, we hypothesized that the binding of Nb2 might tightly seal and stabilize the binding pocket to EL2, thereby slowing hydrolysis of the Schiff base and retinal release. The kinetics of retinal release were measured using multiple spectroscopic techniques (Fig. 5a). First, the rate of cleavage of the deprotonated Schiff base was measured by UV-Vis spectroscopy of hydroxylamine-induced aminolysis over time to evaluate the accessibility of the chromophore-binding pocket in bRho* with and without Nb2 (Fig. 5b). At t = 0 min, the bRho*/Nb2 complex showed a greater Meta I (A478 nm): Meta II (A380 nm) ratio as compared to bRho* alone (Fig. 5b, t = 0 min). Following hydroxylamine

treatment, the decay curve of Meta I-like photointermediates in bRho*/Nb2 plateaus at a 4-times higher value compared to that for Rho* alone, which indicates that Nb binding protects a large population of the bRho* from hydroxylamine-driven Schiff base aminolysis (Fig. 5c). This Nb-dependent protection of bRho* from hydroxylamine-driven retinal release was greater with Nb7, a stronger binder compared to Nb2 (Supplementary Fig. 7a, b).

Next, the hydrolysis of the retinylidene Schiff base in bRho* was measured by LC-MS/MS analysis in the presence or absence of Nb2, using a recently published method[42]. After brief illumination, the level of intact retinylidene Schiff base was monitored over time by reductive Schiff-base trapping and proteolytic digestion followed by LC-MS. The extent of *cis-trans* photoisomerization of the Schiff-base-bound retinal in bRho* was documented by the simultaneous diminution of the N$^\varepsilon$-11-*cis*-retinyl-Lys signal and increase of the N$^\varepsilon$-all-*trans*- retinyl-Lys signal post-illumination (Fig. 5d, dark vs. 0 min, and Supplementary Fig. 7c-h). Notably, the apparent amount of *cis-trans* photoisomerization with Nb2 present was diminished, likely due to either back-isomerization or inner-filter effect by the Meta I-like intermediate state stabilized by Nb2. After a 60 min incubation in the dark post-illumination, there was a sharp decline in N$^\varepsilon$-all-*trans*-retinyl-Lys signal, indicative of hydrolysis of the Schiff base in bRho*, as expected (Fig. 5d, left panel, 0 min vs. 60 min). This hydrolysis was remarkably hindered in the presence of Nb2 (Fig. 5d, right panel, 0 min vs. 60 min), indicating that Nb2 binding protects the Schiff base from hydrolysis. Using the intrinsic Trp fluorescence (F$_{Trp}$)-quenching assay, the release of free retinal from the chromophore binding pocket of bRho can be tracked by monitoring fluorescence recovery at A330nm upon photoactivation. We found that the amount of FTrp recovery for bRho*/Nb2 was relatively lower (Fig. 5e, right panel) than that for bRho* alone (Fig. 5e, left panel), indicating a greater amount of bound retinal content associated with Nb2 complexation. To note, the F$_{Trp}$ of bRho*/Nb2 at t = 0 min is higher than that of bRho* due to the intrinsic Trp fluorescence from Nb2 itself. The relative increase in the Trp fluorescence emission indicating Meta II decay was reduced nearly 3-fold in the presence of Nb2 (Fig. 5f). Collectively, our data confirm that Nb2 binding to the extracellular side of bRho slows the hydrolysis of the Schiff base and the release of retinal upon light activation.

## Nb2 stabilizes and restores the proteostasis of P23H-bOpsin

Rod opsin mutations such as P23H often exhibit inherent protein instability leading to photoreceptor cell degeneration and autosomal dominant retinitis pigmentosa (adRP)[43,44]. The binding of ligands to GPCRs can stabilize protein folding via a pharmacological chaperone effect, often restoring proteostasis of mutant GPCRs, including Rho, δ-opioid, V2 vasopressin, and Frizzled 4 receptors[44–47]. Thus, we hypothesized that the binding of Nb2 could stabilize P23H-bOpsins in an adRP cell model, acting as a chaperone. To test this hypothesis, Nb2 was co-expressed with either WT or P23H-bOpsin in HEK293S cells, followed by western blot analysis of the amount of opsin protein expression. We investigated the stability of retinal-free opsins to determine the effect of Nb2 alone as compared to that of 9-*cis*-retinal

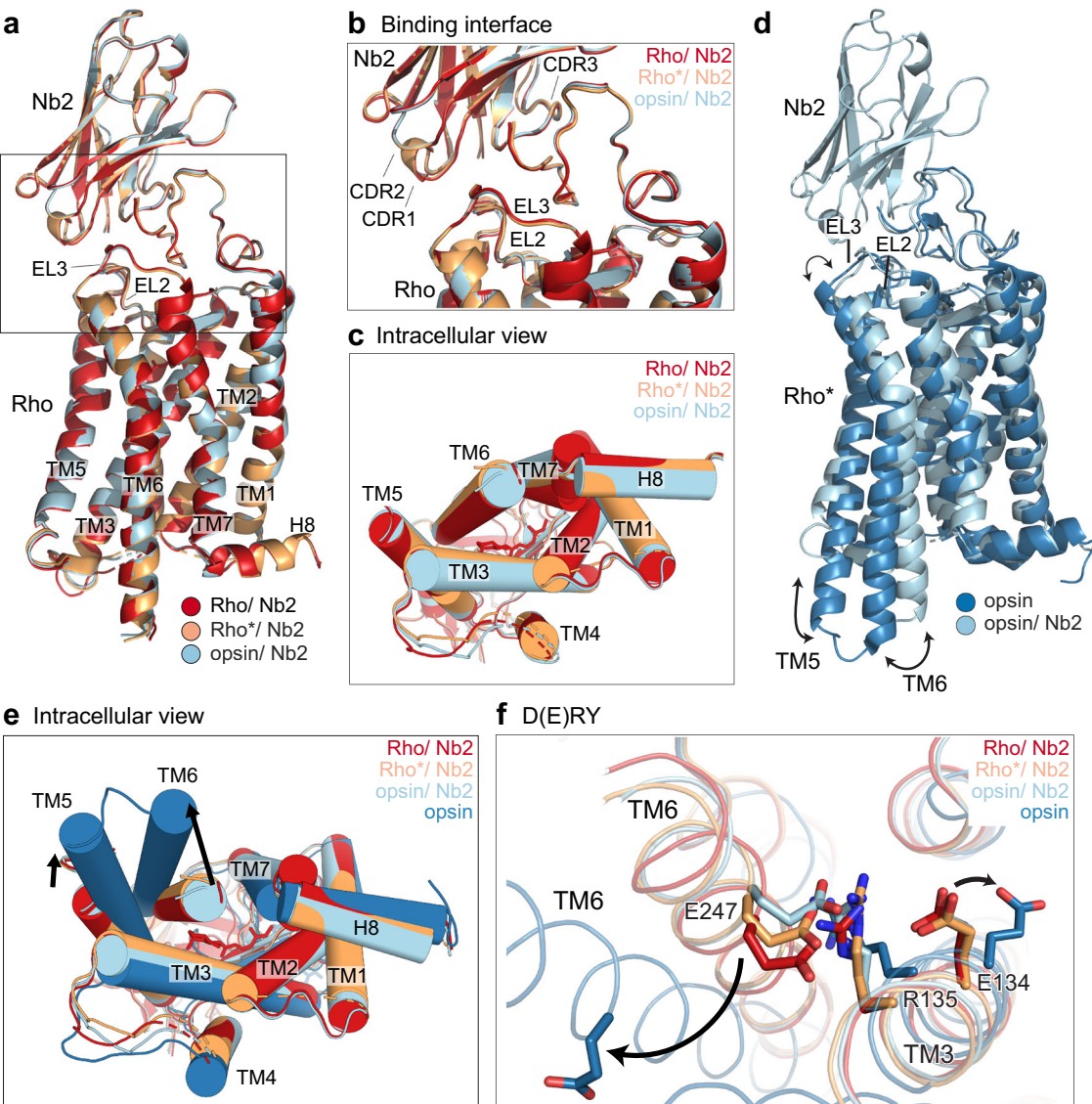

**Fig. 4 | Structural comparisons of bRho/Nb2, bRho\*/Nb2, and apo-opsin/Nb2 with opsin alone. a–c** Superposition of the crystal structures of bRho/Nb2 (red), bRho\*/Nb2 (orange), and apo-opsin/Nb2 (sky blue) complexes at 3.7-Å, 4.25-Å, and 3.7-Å resolution, respectively. Boxed regions in (**a**) are enlarged in (**b**), showing the binding interface between Rho and Nb2. Glycans have been removed for clarity. **d** Structural comparison of apo-opsin/Nb2 (sky blue) with that of apo-opsin (dark blue; PDB ID 5TE3). Black arrows show outward tilting of TM helices 5 and 6.

**e, f** Structural alignment of Rho/Nb2 (red), Rho\*/Nb2 (orange), and apo-opsin/Nb2 (sky blue) with apo-opsin (dark blue). Intracellular view (**e**) shows the outward movements in helices 5 and 6 (indicated by black arrows). D(E)RY motif (**f**) showing the ionic lock between TM3 and TM6 (Thr[251] and 11-*cis*-retinal are not shown). Black arrows indicate movements of Glu[247] on TM6, and Glu[134] on TM3 in the opsin structure.

treatment. Expression of Nb2 significantly increased the opsin levels in both WT-bOpsin and P23H-bOpsin cells by about 1.5-fold and 2-fold, respectively, as compared to the non-binding Nb16 control (Fig. 6a, b). The increase in WT or P23H-bOpsin by Nb2 co-expression was comparable to the treatment with 9-*cis*-retinal. Treatment of either WT or P23H-bOpsin cells with exogenous Nb2 did not change the expression levels of opsins (Supplementary Fig. 8a, b), suggesting Nb2 is required intracellularly at the ER to protect against loss of opsin proteostasis.

Next, we tested whether the Nb2-rescued P23H-bOpsin exhibits proper localization to the plasma membrane. WT- or P23H-bOpsin cells with co-expression of either Nb2 or Nb16 were subsequently stained with anti-Rho 1D4 antibody (Fig. 6c). Intracellular expression of Nb2 increased the level of opsin on the plasma membrane in both WT- and P23H-bOpsin cells compared to Nb16-expressing negative control cells. Interestingly, the Nb2- expressing cells appeared to have extended cell morphology with increased cell surfaces compared to 9-*cis*-

retinal treated cells, which also showed enhanced plasma membrane-localized P23H-bOpsin. Such findings suggest that localization of WT or P23H-bOpsins to the plasma membrane likely occurred to a greater extent as a result of increased 1D4 immunostaining at the membrane with co-expression of Nb2. The rounder morphology of 9-*cis*-retinal-treated cells may relate to non-specific oxidative stress caused by the retinaldehyde treatment.

The effect of Nb2 on the hydrolysis of the retinylidene Schiff base was also explored for both WT- and P23H- bOpsin cells by LC-MS. P23H-bOpsin showed a complete *cis-trans* conversion (Fig. 6f, left panel, dark vs. 0 min) and decline in all-trans-retinyl-Lys signal over time (Fig. 6f, left panel, 60 min), indicating that the mutant opsin photochemistry and hydrolysis were intact, as was observed also for WT-bOpsin (Fig. 6d, left panel). In the presence of Nb2, the extent of hydrolysis was greatly attenuated as shown by the relatively larger signal of all- trans-retinyl-Lys at 60 min post-illumination with Nb2 in

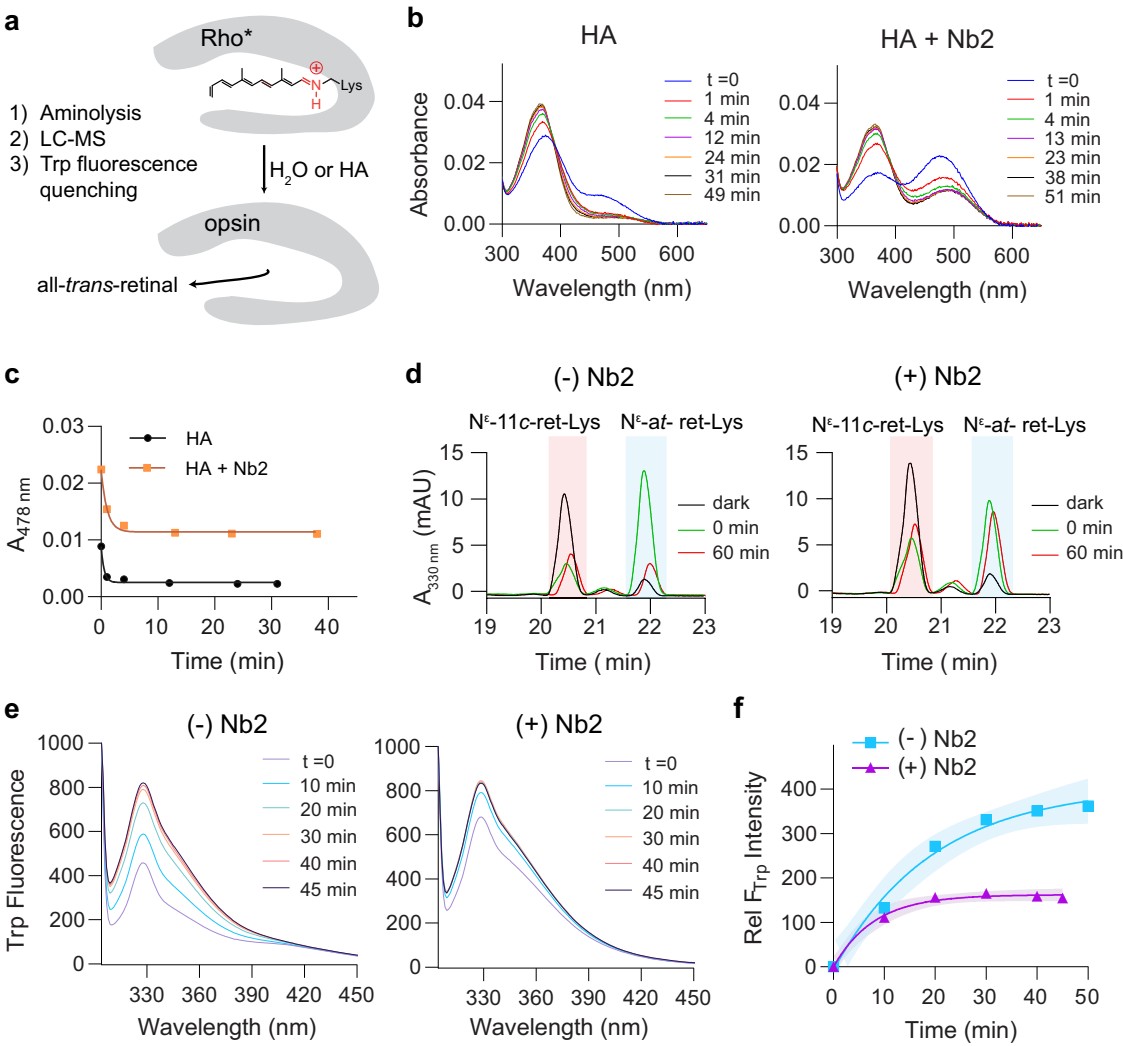

**Fig. 5 | Hydrolysis of the Schiff base and release of retinal following photo-activation is slowed by bRho- Nb2 complex formation. a** Experimental scheme for testing the hydrolysis of the Schiff base and release of retinal. HA stands for hydroxylamine. **b** UV-Vis spectra of bRho* and of bRho*/Nb2 complexes were measured after the addition of 5 mM HA. **c** A478nm values from spectra in (**b**) were plotted using non-linear one-phase decay. **d** LC-MS chromatogram of pronase proteolysis of NaBH₄-treated bRho in DDM, with or without Nb2 bound. The peak at ~20.4 min corresponds to Nᵉ-11-*cis*-retinyl-Lys (pink), whereas the peak at ~21.9 min corresponds to Nᵉ-all- *trans*-retinyl-Lys (blue). The peak between the 11-*cis* and all-*trans* isomer peaks corresponds to Nε-9-*cis*-retinyl- Lys ($\lambda_{max}$ = 325 nm)[42], a

byproduct of photoisomerization of early protonated all-*trans* retinylidene photointermediates[93, 94]. **e** Fluorescence emission spectra of intrinsic tryptophan residues of bRho in the absence (left) or presence (right) of Nb2, at 298 K (pH 7.0), at indicated time points post-illumination. **f** Relative tryptophan fluorescence values at 330 nm at indicated time points were measured, normalized to t = 0 min, plotted, and fit to a single exponential association kinetic curve with R²= 0.9881 (Nb-) and R²= 0.9920 (Nb+). Shaded areas indicate the 95% confidence interval. The experiment was carried out one time. Source data for panels **b**-**f** are provided as a Source Data file.

both WT- and P23H-bOpsin cells (Fig. 6d, f, right panel), as compared to without Nb2 (Fig. 6d, f, left panel). We also confirmed the decreased hydrolysis by analyzing the release of free retinal in the supernatant by LC-MS (Fig. 6e, g, Supplementary Fig. 8c, d). As we expected, the amount of hydrolyzed retinal in both WT- and P23H-bOpsin was lower with Nb2 after 60 min incubation post-illumination, compared to that without Nb2 (Fig. 6g, cyan, 0 min vs. 60 min).

## Discussion

Nbs are revolutionizing the field of GPCR structural biology because their small size and structural characteristics, such as a protruding paratope, allow them to bind small protein cavities and epitopes close to the cell membrane. Several Nbs stabilizing GPCR conformations have been characterized structurally and have been shown to bind to the intracellular domain of GPCRs[48–50], where the larger conformational changes occur upon GPCR activation. Although several nanobodies modulating the activity of class-A GPCRs by targeting their

extracellular domain have been developed, there is little information on the structural basis for their effect. Recently a nanobody was shown to have an orthosteric antagonistic activity towards APJ[51]. We show that Nb2 exerts its allosteric antagonist activity on Rho* by binding to its extracellular domain and preventing the otherwise small conformational changes associated with full activation despite the cognate agonist (all-*trans*-retinal) being present in the binding pocket.

To date, more than 150 cryo-EM and crystal structures of GPCRs have been determined with the aid of Nbs[51]. With two exceptions (OX2R[52] and APJ[53]), in these structures of class-A GPCRs the Nb is bound on their intracellular side, in many cases not directly interacting with the receptor. However, in several cases, the Nb reaches inside the 7-TM bundle in a manner like G proteins, stabilizing the active conformation of the GPCR. Examples include the β₂-adrenergic receptor (PDB 3P0G, 6N48, 4QKX), β₁-adrenergic receptor (PDB 6IBL), μ- opioid receptor (PDB 5C1M), and M2 muscarinic acetylcholine receptor (PDB 4MQS). Here, we show the crystal structures of a class-A GPCR

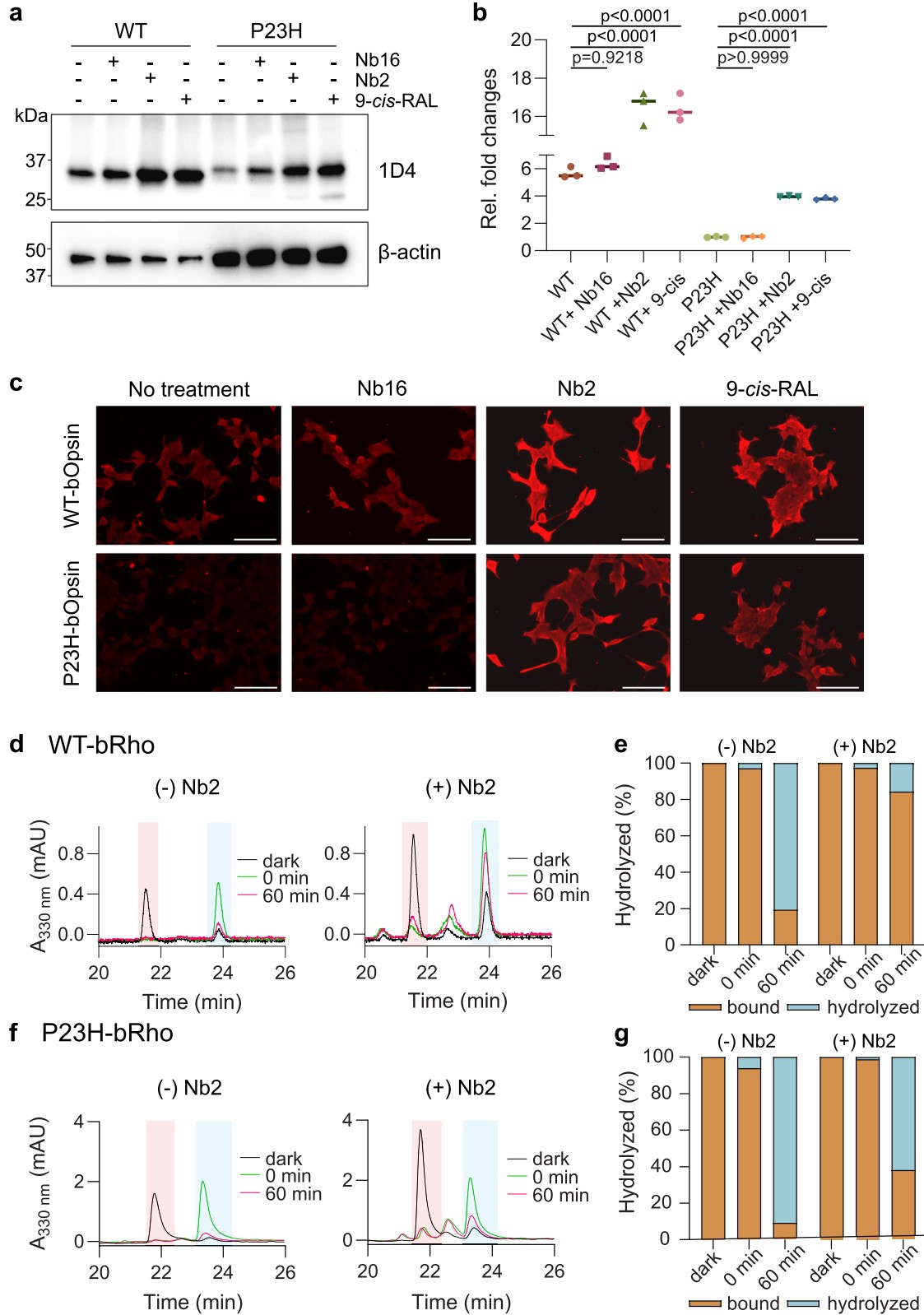

complexed with an extracellular Nb that allosterically modulates the receptor.

The bRho/Nb2 crystal structure revealed that several critical interactions involving residues in the N-terminus and EL2 stabilize the complex. In class-A GPCRs, EL2 and N-terminus can be involved in ligand recognition, capping of the ligand binding pocket and in the activation process[54–57]. In rhodopsin's ground state, the N- terminus and EL2 serve as a cap on the retinal-binding site, preventing the access of hydrophilic molecules to the protonated retinylidene Schiff base[1,41]. EL2 contains Glu[181] which acts as the counterion for the protonated Schiff base in the Meta I state, instead of Glu[113] [(3.28)] in the ground state[58,59]. In class-A GPCRs, EL2 is the largest and most divergent of the three extracellular loops, and it can contribute to ligand selectivity[54]. In addition, a disulfide bridge between Cys[187] in EL2 and Cys[110] [(3.25)],

**Fig. 6 | Nb2 restores the proteostasis of bovine P23H-opsin. a, b** Nb2 or non-binding Nb16 were co- expressed intracellularly in WT-bOpsin-expressing or P23H-bOpsin-expressing HEK293 cells, followed by western blot analysis with anti-Rho 1D4 antibody. Untreated WT- and P23H-bOpsin-expressing cells were used as a negative control. Overnight treatment with 2.5 μM of 9-cis-retinal was used as a positive control for Rho stabilization. 5 μg of WT-bOpsin and 15 μg of P23H-bOpsin protein lysates were loaded per lane. **b** Quantification results show relative fold changes against P23H in levels of 1D4 expression, calculated from panel (**a**) after normalization to β-actin levels. The data are expressed as the means (horizontal bars) together with individual data points (filled shapes, $n = 3$ biologically individual samples). One-way ANOVA was performed to determine statistically significant differences. Adjusted $P$-values are indicated. **c** Fluorescence images of 1D4-Alexa647- immunostained WT-bOpsin cells or P23H-bOpsin cells, with co-expression of Nb2 or Nb16 (negative control). 9- cis-retinal treatments were used as

positive controls for the rescue of misfolded P23H-bOpsin. Images are representative of 3 biologically independent experiments. Scale bars, 250 μm. **d**–**g** LC-MS chromatogram of WT-bRho (**d**) or P23H-bRho (**f**), purified from HEK293S cells with or without Nb2, comparing before ("dark", black curve) and after brief light exposure at 0 min ("0 min", green curve), and after 1 h dark incubation post illumination ("60 min", red curve). The peak at -21.6 min corresponds to $N^\varepsilon$-11-cis-retinyl-Lys (pink), whereas the peak at -23.5 min corresponds to $N^\varepsilon$-all-trans-retinyl-Lys (blue). The peak between the 11-cis and all-trans isomer peaks corresponds to $N\varepsilon$-9-cis-retinyl-Lys ($\lambda_{max} = 325$ nm)[42], a byproduct of photoisomerization of early protonated all- trans retinylidene photointermediates[93, 94]. **e, g** Supernatants from (**d**) and (**f**) were analyzed by LC-MS for hydrolyzed free retinal (light blue), compared to bound chromophore (orange) in the membrane pellets. Source data for panels **a**–**g** are provided as a Source Data file.

conserved in GPCRs, controls the movements between TM3, TM5 and EL2 upon activation. These facts would explain why locking the N-terminus and EL2 in the ground-state-like conformation by a Nb would prevent Rho* from progressing to the fully activated Meta II state. Moreover, P23H-mutant-opsin with intrinsic protein instability was rescued from misfolding and degradation by Nbs binding, suggesting that Nb2 can act as a molecular chaperone like 9-cis-retinal[60].

With a few exceptions, GPCRs carry at least one putative N-glycosylation site in their N-terminal ectodomain or extracellular loops[61,62], and more than 350 GPCRs have been predicted to be O-glycosylated[63]. Consequently, we anticipate that a significant fraction of therapeutic antibodies binding native GPCRs will interact with the glycans. However, GPCR constructs used for structural studies are often deglycosylated and, therefore, there is a lack of structural information on the possible role of glycans in GPCR recognition and regulation by antibodies. In addition, antibodies raised against deglycosylated GPCRs might not bind the native GPCR form. Our work shows an example of antibody/glycan interaction in a class-A GPCR, providing the structural basis to understand the role of glycans in antibody/GPCR recognition and conformational modulation. We show out that CDR2, CDR3, and, surprisingly, the FR2 loop of Nb2 maintain extensive interactions with Rho's native glycans in the three crystal structures. We also found that hyperglycosylation or removal of any of the two N-glycans of bRho reduced or abolished Nb2 binding. These results reinforce the concept that native GPCR post-translational modifications like glycosylation, which often are overlooked, need to be considered in the development of biologics[64].

Rho's early photoactivation states are not completely characterized, and biophysical and structural characterization of Batho, Lumi, and Meta I have been performed by FTIR or crystallography by trapping these intermediates at low or ultralow temperatures[35,37,65]. Recently, the conformational changes of rhodopsin from its ground-state up to the Batho state were determined at room temperature by ultrafast, time-resolved serial crystallography[66]. In the present work, we have obtained a set of Nbs that stabilize photoactivated Rho at the Lumi or Meta I state at near-physiological temperatures, as demonstrated by UV-Vis and FTIR spectroscopy and X-ray crystallography. In addition, data from intrinsic Trp fluorescence and UV-Vis spectroscopy in the context of hydroxylamine or NaBH$_4$-reduction experiments demonstrate that Nb binding to bRho prevents or slows down the hydrolysis of the Schiff base and subsequent release of all-trans-retinal. Importantly, UV-Vis spectroscopy revealed that the addition of Nbs to bRho* (mainly in the Meta II state) resulted in a shift of the equilibrium towards a Meta I-like state, a transition that involves re-protonation of the Schiff base and deprotonation of its counterion Glu$^{113}$ (3.28), re-formation of the ionic lock, and significant conformational changes, mainly involving TM5 and TM6[67]. This equilibrium shift likely occurs by Nb2 binding to Meta I, which is present at a low population in the sample, rather than binding directly to Meta II, due to Nb antigen recognition via conformational selection[48]. In a similar manner, Nb2

forms crystals with opsin, stabilizing a Meta I-like conformation. This observation is in line with previous FTIR studies suggesting that opsin is in equilibrium between an active opsin (Ops*) state resembling Meta II, and an inactive state (Ops) resembling a Meta I state[68]. It is noted that UV-vis spectroscopy revealed that Nb binding to detergent-solubilized bRho* shifts the equilibrium from Meta-II to a Meta-I-like state, whereas FTIR experiments showed that Nb binding to bRho in membranes stops the conformational transition triggered by photo-activation mainly at the Lumi state.

There is a growing interest in the invention of therapeutic biologics, including mAbs, that modulate GPCRs via allosteric binding. Such interest is driven by the favorable safety profiles and typically longer half-lives of biologics as compared to small molecules[69,70]. Approximately 35% of FDA-approved drugs target GPCRs, yet there are only five FDA-approved monoclonal antibodies (mAbs) targeting GPCRs and their ligands[71]. Among them, erenumab is the only mAb that disrupts ligand-receptor interactions by directly binding to the calcitonin-gene-related peptide (CGRP) receptor[72]. Understanding the structural basis of GPCR activation might aid in designing and developing therapeutic antibody drugs[73]. Here we show a series of Nbs that bind to the extracellular surfaces of rhodopsin and act as allosteric modulators, providing an insight into structure-guided biological drug development. This Nb technology could be further developed to stabilize different GPCR conformational states, providing therapeutic diversity.

In summary, we discovered and characterized a series of Nbs that enabled us to draw the following conclusions: i) the Nbs act as allosteric modulators by binding to the extracellular surface of bRho and stabilizing the photoactivated receptor in a ground-state-like, inactive structure. This effect was demonstrated under various conditions, including detergent solution, ROS membranes, detergent-solubilized membranes, and in crystallo; ii) the binding of Nb significantly reduce the accessibility of solvent to the binding pocket, slowing down the hydrolysis of the retinylidene Schiff base of bRho* and the release of all-trans-retinal; iii) when co-expressed with the misfolding-prone P23H-bOpsin mutant, Nb2 acts as a chaperone to partially restore proteostasis; and iv) Nb2 facilitates crystallogenesis of bRho and opsin at near physiological pH. The Nb-stabilization strategy shown here is likely to facilitate a long-term goal in the field by structurally resolving the key intermediate states of Rho, from the inactive state to the fully ligand-activated state, as well as the apoprotein and other states such as meta III. In addition, this research opens the door to the development of biological allosteric modulators that bind to the extracellular domain of other GPCRs and stabilize different activation states.

## Methods
### Animal approvals
Ten, both male and female, 6-8-week-old C57BL6/J mice were purchased from the Jackson Laboratory (Jackson Laboratory; Bar Harbor;

strain # 000664) for preparing mouse rod outer segments. Mice were housed in the vivarium at the University of California, Irvine, where they were maintained on a normal mouse chow diet and a 12 h/12 h light (<10 lux) per dark cycle. Breeding and experimental rooms were maintained at an ambient temperature of 20–26 °C and a humidity of 40–60%. Mouse experiments were approved by the Institutional Animal Care and Use Committees (IACUC) of the University of California, Irvine, and the VA Long Beach Health Care System and were conducted in accordance with the Association for Research in Vision and Ophthalmology Statement for the Use of Animals in Ophthalmic and Visual Research. One female, 6-years-and-7-months-old *Llama glama* was used for vaccination experiments, executed in strict accordance with good animal practices, following the EU animal welfare legislation and with approval of the local ethical committee (Ethical Committee for use of laboratory animals of the Vrije Universiteit Brussel).

## Reagents
Anti-β-actin (catalog no. 4970) was purchased from Cell Signaling Technology (MA, USA). Mouse monoclonal anti-Rho 1D4 antibody was purified from hybridoma cells[74–76]. Alexa Fluor 647-conjugated goat anti-mouse IgG antibody (catalog no. A21236) was from Invitrogen (CA, USA). DAPI fluorescent dye (catalog no. 62248) was purchased from Thermo Fisher (MA, USA). All detergents were purchased from Anatrace (OH, USA). Hydroxylamine hydrochloride (catalog no. 159417) was purchased from Sigma-Aldrich (MI, USA). 9-*cis*-retinal and 11-*cis*-retinal were produced by photoisomerization of all-*trans*-retinal using 420 nm light and purified by normal-phase HPLC using a Phenomenex Luna silica column (catalog no. 00G-4091-P0-AX) and an isocratic gradient of 10% ethyl acetate in hexanes[77,78].

## Nb generation and discovery
ROS were isolated from dark-adapted bovine retinas by step-sucrose density gradient centrifugation following established protocols and washed several times with isotonic and hypotonic buffers[79,80]. These samples were kept in the dark as much as possible and were used as an immunogen to immunize a llama. Nbs against ROS were obtained after phage display selection, using established protocols[81]. Briefly, a llama was immunized 6 times with bROS. Peripheral blood lymphocytes were collected after the final boost and RNA was purified to amplify the ORF of the Nbs to generate a phage display library. bROS samples were diluted in 5 mM HEPES pH 7.5, 0.1 mM EDTA to adsorb them to the plate. After biopanning and an ELISA, 18 different families, classified according to the sequences of the third complementarity determining region (CDR3), were retained for further analysis.

## Nb expression and purification
Nbs were expressed and purified according to a previously published protocol[81]. Briefly, Nbs bearing a C-terminal His-tag were transformed in *E. coli* WK6 (Su⁻) cells. Small-scale cultures of 20 ml were grown in LB media containing 2% glucose, 1 mM MgCl$_2$, and 50 µg ml⁻¹ ampicillin. Large-scale cultures of 2 L were grown to $A_{600\,nm}$ = 0.6 – 0.7 at 37 °C in TB medium containing 0.1% glucose, 1 mM MgCl$_2$, and 50 µg ml⁻¹ ampicillin. Cultures were induced with 1 mM IPTG and grown overnight at 28 °C. Cells were harvested by centrifugation at 7000 × $g$ for 10 min at 4 °C. The periplasmic fraction was extracted by resuspending the cells in an ice-cold TES buffer containing 0.2 M Tris, pH 8.0, 0.5 mM EDTA, and 0.5 M sucrose. Following a 1 h incubation at 4 °C, cells were supplemented with ¼-strength TES buffer to achieve a final buffer composition of 0.1 M Tris, pH 8.0, 0.25 mM EDTA, and 0.25 M sucrose. After a 45 min incubation at 4 °C, cells were removed by centrifugation at 8000 × $g$ for 30 min at 4 °C. The periplasmic extracts were incubated with cOmplete His-Tag purification resin (catalog no.05893801001, Roche-Millipore Sigma, MA, USA) on a rotary shaker at 4 °C for 2 hr. The extract-resin mixture was added onto a polyethylene filter gravity column and washed with forty resin

volumes of wash buffer (50 mM HEPES, 300 mM NaCl, 3 mM imidazole) and eluted with 300 mM imidazole. Nbs were dialyzed against 10 mM HEPES and 50 mM NaCl overnight and concentrated to ~50 mg ml⁻¹ using a 10 kDa Amicon ultracentrifugal filter (catalog no. UFC901096, Millipore Sigma).

## Blue native polyacrylamide gel electrophoresis (BN-PAGE) assay
BN-PAGE was performed in a XCell SureLock Mini-Cell electrophoresis system (catalog no. EI001, Invitrogen), using NativePAGE mini gels (catalog no. BN1002, Invitrogen) in the dark or light. 3 µg of 1D4-purified Rho was incubated with 2-fold incremental amount of Nbs in buffer containing 20 mM HEPES, 20 mM NaCl and 0.05% DDM for 2 h in the dark and supplemented with sample buffer (12.5% glycerol, 0.025% Ponceau S, pH 7.0) before loading. Unstained protein standards (catalog no. LC0725, Invitrogen) were used for a protein ladder. Cathode and anode running buffers were used in inner and outer chambers, respectively; buffer compositions were 50 mM Bis-Tris propane (BTP)/50 mM Tricine (pH 6.8) /0.02% Coomassie G-250 and 50 mM BTP/ mM Tricine (pH 6.8), respectively. Electrophoresis was run on ice, starting at 150 V for 2 h and continued at 250 V for 2 h. The gels were stained with Coomassie blue stain (catalog no. 50-109-1216, Anatrace).

## Alanine scanning mutagenesis
Nb2 mutants were generated via site-directed mutagenesis using a Takara In-Fusion Snap Assembly system (catalog no. 638951, Takara Bio, Shiga, Japan) and purified as described above. For mROS, 6-8 retinas from 6-8-week-old wild-type C57B6/J mice (Jackson Laboratory, ME, USA) were isolated and placed in ice-cold Ringer's buffer (10 mM HEPES, 130 mM NaCl, 3.6 mM KCl, 12 mM MgCl$_2$, 0.02 mM EDTA, pH 7.4). Isolated retinas were suspended in 0.08 ml of 8% (v/v) OptiPrep (catalog no. AXS1114542, Cosmo Bio USA, Inc, CA, USA) in Ringer's buffer and vortexed at maximum speed for 1 min. The samples were centrifuged at 250 × g for 1 min at 4 °C and the supernatant containing ROS was collected. The vortexing and sedimentation sequence was repeated six times. For bROS, one bovine retina was isolated, finely chopped with dissection scissors (Fine Science Tools, B.C., Canada), and placed in 5 ml of 8% (v/v) OptiPrep in Ringer's buffer. The vortexing and sedimentation sequence was repeated six times.

## Co-immunoprecipitation
For the Ni-NTA affinity chromatography assay with WT or mutant Nbs, 100 µg of isolated mROS or bROS were incubated with 50 µl of cOmplete His-Tag purification resin and 40 µg of Nb in binding buffer (50 mM HEPES, pH 7.4, 250 mM NaCl, 20 mM CHAPS, and 1 mM DDM) for 2 h on a shaker at 4 °C. The ROS-Nb-resin mixture was loaded into a polyethylene filter gravity column and washed with 40 resin volumes of wash buffer (50 mM HEPES, 300 mM NaCl, 20 mM CHAPS, and 3 mM imidazole, pH), and eluted with the same buffer containing 300 mM imidazole, pH 7.4. Co-immunoprecipitation assays were carried out three times with similar results.

## Stable cell line generation for recombinant Rho
HEK293S cell lines expressing bovine WT-bOpsin (NCBI gene ID 509933), P23H-bOpsin, WT-mOpsin, mouse/bovine EL2-opsin, N2Q bOpsin, or N15Q bOpsin were each generated by transduction of HEK293S cells with the appropriate retrovirus construct obtained from Phoenix-AMPHO packaging cells (catalog no. CRL-3213, ATCC, VA, USA), followed by fluorescence-activated cell sorting (FACS) for GFP-positive populations, using a BD FACSAria Fusion cell sorter. Phoenix-Ampho cells were transfected with 10 µg of pMXs-bWT-opsin-IRES-GFP, pMXs-P23H-bOpsin-IRES-GFP, pMXs-WT- bOpsin-IRES-GFP, pMXs-m/b-EL2-opsin-IRES-GFP, pMXs-N2Q-bOpsin-IRES-GFP, or pMXs-N15Q-bOpsin- IRES-GFP, using 30 µl of GeneJuice transfection reagent (catalog no. 70967, Millipore Sigma). The transduced Phoenix-Ampho

cells were treated with 10 mM sodium butyrate (Catalog no. B5887, Sigma-Aldrich) for 8 h and the conditioned medium was collected 48 h after the transfection. The stable cell lines were cultured in growth medium composed of Dulbecco's modified Eagle's medium/F12, pH 7.2, with 4 mM L-glutamine, 4500 mg L$^{-1}$ glucose, and 110 mg L$^{-1}$ sodium pyruvate, supplemented with 10% heat-inactivated fetal bovine serum (catalog no. A3840001, Thermo Fisher), 100 units mL$^{-1}$ penicillin, and 100 units mL$^{-1}$ streptomycin. Cells were maintained at 37 °C in 5% CO$_2$. WT-bOpsin and WT-mOpsin (NCBI gene ID 212541) cDNA were purchased from Gene Universal (DE, USA). P23H-bOpsin, N2Q bOpsin, and N15Q bOpsin-expressing plasmids were generated by site-directed mutagenesis, using a Takara In-Fusion Snap Assembly system, with the WT plasmids as a template. The Nb2-expressing plasmid was generated by subcloning the Nb2 sequence from pMESY4 into the pMXs-IRES-GFP retroviral expression vector (catalog no. RTV-013, Cell Biolabs, Inc., CA, USA).

## Purification of recombinant Rho by immunoaffinity chromatography

HEK293 cells expressing WT-bOpsin, P23H-bOpsin, N2Q-bOpsin, or N15Q-bOpsin were cultured in 1 x p150 mm culture dishes and then pelleted down for 1D4-affinity chromatography. Immunoaffinity 1D4 resin was prepared by conjugating purified, anti-Rho antibody (1D4) to CNBr-activated Sepharose 4B beads (catalog no. 17-0430-01, Cytiva, MA, USA). Pelleted cells were washed with PBS and harvested in hypotonic buffer (10 mM Tris, pH 7.4, 1 mM MgCl$_2$) supplemented with Roche cOmplete proteinase inhibitor cocktail (catalog no. 11697498001, Roche-Sigma-Aldrich), followed by centrifugation at 10,000 × g for 10 min at 4 °C. Then, the pellets were homogenized by passing through a 23-gauge syringe in a hypertonic buffer (10 mM Tris, pH 7.4, 1 mM MgCl$_2$, and 1 M NaCl), followed by centrifugation. The membrane-enriched pellets were solubilized for 2 h in the cold room in an isotonic buffer (10 mM Tris, pH 7.4, 1 mM MgCl$_2$ and 0.25 M NaCl), supplemented with 10% DDM. Then, 50 μl of 1D4-resin was added to the sample, followed by 1 h incubation in the cold room. Rho-1D4-resin mixture was loaded into a centrifuge column (catalog no. 89897, Pierce-Thermo Scientific), washed in a buffer containing 10 mM HEPES, 0.25 M NaCl, and 10% DDM, and eluted with C-terminal nonapeptide (synthesized by GenScript, NJ, USA).

## Western blotting

Cells were lysed with PBS containing 0.1% Triton X-100 and protease inhibitor cocktail (Roche), followed by sonication for 3 sec. Cell lysates were centrifuged at 10,000 × g for 10 min at 4 °C and the supernatant was collected for determining protein concentration by BCA assay (catalog no. 23227, Pierce-Thermo Scientific). Protein lysates were loaded, separated by SDS–PAGE (catalog no. 4561096, Bio-Rad, CA, USA), transferred to PVDF membranes (catalog no. L00686, eBlot, GenScript), blocked with 5% milk/TBST, and probed with antibodies at 4 °C overnight. Uncropped blots are provided in the Source Data.

## Surface Plasmon Resonance (SPR) Measurements

SPR measurements were performed using the OpenSPR Rev 4 (Nicoya, ON, Canada) at 20 °C under ambient lighting conditions. Affinity-purified Nb2 (25 ng ml$^{-1}$) was diluted in running buffer (50 mM HEPES, pH 7.4, 0.15 M NaCl, and 0.04 mM LMNG) and immobilized on a carboxyl sensor chip using the Amine Coupling Kit (catalog no. AMINE-10, Nicoya), according to the manufacturer's instructions. Interaction of 1D4-purified bRho with the immobilized Nb2 was tested by application of bRho in varying concentrations ranging from 62.5 to 1000 nM in running buffer (50 mM HEPES, pH 7.4, 150 mM NaCl, and 0.04 mM LMNG). The association and dissociation experiments were performed at a flow rate of 20 μl min$^{-1}$ with a contact time of 235 sec and dissociation time of 450 s. Data processing and analysis were performed using TraceDrawer software (Ridgeview Instruments, Uppsala,

Sweden) to obtain $k_{on}$, $k_{off}$, and $K_D$ constants based on the 1:1 Langmuir binding model.

## UV-Visible spectroscopy

For UV-visible spectroscopy concentrated samples of photoactivated Rho in complex with Nbs were prepared in a manner like those for crystallography, except that β-mercaptoethanol was omitted and DDM was used as a final solubilization detergent.

To study the Nb-induced shift in equilibrium from Meta-II to Meta-I, bRho solubilized in DDM was photoactivated on ice for 30 sec with 505-nm or 535-nm fiber light (625 μW), and the resulting bRho* was mixed with the appropriate Nbs at different pHs. The final concentrations in the samples were: 6 mg mL$^{-1}$ bRho* (150 μM), 300 μM Nbs, 0.1 M BTP, 0.15 M NaCl, 5 mM EDTA. The samples were incubated on ice in the dark to reach equilibrium. Then, an aliquot of each sample was diluted 100 times in the same buffer containing 1 mM DDM and the absorption spectrum was measured immediately.

To study the protection by Nb2 of bRho* against hydroxylamine (NH$_2$OH, HA)-mediated aminolysis, photoactivation of bRho was achieved with 565-nm diode fiber light. After the addition of bRho* to Nbs and subsequent 4 h incubation on ice, each sample was diluted 200 times and the absorption spectrum was measured. Immediately, 5 mM hydroxylamine was added from a 0.5 M stock solution, and changes in the spectrum at room temperature were measured over time. In all cases, samples of Rho* without Nb were prepared as controls.

For the long-term stability of Nb2/bRho complexes, bRho was purified with immobilized 1D4 antibody in 1 mM DDM, 20 mM HEPES, pH 7.4, 0.15 M NaCl. Then, Nb2 was added at a 2-times molar concentration over that of Rho.

## Rho purification for crystallization

A small excess of 11-cis-retinal was added to ROS to regenerate Rho from apo-opsin, which can account for 10-65% of the total opsin in the ROS, depending on the source of bovine retinas. After 2 h of incubation at room temperature, the ROS were stored at −80 °C.

For bRho purification for crystallization, ROS membranes (10-20 mg mL$^{-1}$ Rho) were solubilized by zinc/alkyl- glucoside extraction[82]. Specifically, ROS were solubilized with a 3:1 (w/w) mixture of 4-cyclohexyl-1-butyl-β-D-glucoside (Cyglu-4, catalog no. C324G, Anatrace) and n-heptane-1,2,3-triol (HPTO, catalog no. 51845, Sigma-Aldrich) at a ratio Cyglu-4/bRho of ~2.1/1 (w/w). Additional components of the solubilization buffer were 50 mM Na acetate, pH 6.0 and 120 mM ZnCl$_2$. After 4 h of incubation at room temperature, the solubilized bRho was recovered by centrifugation. To remove ZnCl$_2$ and detergent, the sample was dialyzed overnight at 4 °C against 2 mM Tricine, pH 7.5, 1 mM EDTA. The insoluble bRho-lipid mixture was resuspended with ~1.5 mL of 20 mM EDTA in 5 mM Tricine, pH 7.8, followed by centrifugation at 4 °C. The pellet was resuspended in 2 mM Tricine pH 7.8, 10 mM EDTA, 10 mM β-mercaptoethanol; and solubilized with 10% Cyglu-4 at a ratio Cyglu-4/bRho of ~2.3:1 (w/w) to obtain a bRho concentration of 12–15 mg ml$^{-1}$. Nb2 was added to bRho at an approximate molar ratio bRho/Nb2 of 2:3, and the sample was filtered through a 0.22 μm centrifugal filter to obtain a final concentration of bRho of ~7 mg mL$^{-1}$.

## Crystallization of bRho/Nb2 complex

Crystallization of the bRho/Nb2 complex was carried out by vapor diffusion in sitting-drop 96-well plates by mixing equal volumes of bRho/Nb2 mixture and crystallization buffer (0.1 M Tricine pH 8.0, 30% (v/v) PEG 600). The bRho/Nb2 sample was supplemented with several additives: 1.2 mM allosteric ligand F3215-0002 (Life Chemicals, ON, Canada)[83], 2% (w/v) IPTG, and 50 mM phenol. The plate was set up in a dark room, incubated at 22 °C for three days, and then placed at 4 °C. After two weeks, crystals were harvested with cryoloop (Mitegen, NY,

USA) under the microscope with red filters and flash-frozen in liquid nitrogen.

## Crystallization of bRho*/Nb2 complex

To obtain bRho/Nb2 crystals, the bRho/Nb2 sample was equilibrated against 0.1 M Tricine pH 7.8 and 25.5% (v/v) PEG 600. Then, the crystals were illuminated with intense green light (~200 μW at 500 nm) for 6 min at 4 °C, before harvesting also under green light in a 4 °C cold room. Color changes of the crystals from red to orange were observed after illumination with green or white light.

## Crystallization of opsin/Nb2 complex

A bRho/Nb2 sample was bleached under intense white light for 5 min in the presence of 10 mM hydroxylamine, and the crystallization plate was set up under normal light conditions. The crystallization buffer was 0.1 M Tricine pH 8.16 plus 25% (v/v) PEG 600, supplemented with 2% IPTG. The plate was initially incubated at 4 °C for 18 days. Then, PEG 600 was added to the reservoir buffer to reach a concentration of 30% (v/v) and the drop was allowed to equilibrate for an extra month at 4 °C before harvesting the crystals in a 4 °C cold room.

## X-ray-diffraction data collection and processing and structure determination

Diffraction data for bRho/Nb2 and bRho*/Nb2 crystals were collected at the APS NECAT and SSRL 12-2 and 12-1 beamlines under cryogenic temperatures with the hutch darkened and the crystals illuminated by dim red light. Crystals of opsin/Nb2 were collected under the normal lighting of the experimental end-station. Diffraction data were processed and reduced using the XDS package[84]. Data collection statistics are shown in Table S1. Initial phases for an opsin/Nb2 crystal diffracting ~5 Å were obtained by molecular replacement using a deposited Rho structure (PDB accession code: 1L9H, chain A) as a search model within the BALBES structure solution pipeline[85] of the CCP4 online webserver. A well-defined partial solution with two Rho molecules per asymmetric unit was obtained in space group $P3_121$. This model was then used as a fixed solution for a second round of molecular replacement in Phaser[86], using a deposited Nb structure as a search model (PDB accession code: 4BEL, chain C). Two copies of the Nbs were found in the asymmetric unit. The 2Rho-2Nb2 structure was then subjected to alternating rounds of real space model building, updating, and refinement in Coot[87,88], followed by reciprocal space refinement in REFMAC[89]. Side chains were modeled taking into consideration the density map and geometric plausibility, as well as their position in a higher resolution rhodopsin structure (PDB accession code: 1U19 [https://doi.org/10.2210/pdb4BEL/pdb], chain A). Subsequent isomorphous bRho/Nb2, bRho*/Nb2, and opsin/Nb2 crystals were solved by direct rigid-body refinement in REFMAC and further refined as described above. Structures were validated using the Molprobity[90] and wwPDB[91] webservers as well as Privateer[92]. Structure refinement and validation statistics are shown in Table S1.

## ATR-FTIR spectroscopy

Light-induced structural changes of bRho/Nbs complexes were measured by attenuated total reflection (ATR)- Fourier transform infrared (FTIR) spectroscopy. The protein sample (3 μL of bRho at a concentration of 10 μM) was placed on the surface of a silicon ATR prism, which generates three internal total reflections (Smiths Detection, London, UK), and gently dried. The sample was rehydrated with the buffer solution (0.1 M NaCl, 0.2 NaH$_2$PO$_4$, pH 7.0) including Nbs at a final concentration of 50 μM, 5 times higher than the concentration of bRho. The rehydrated film sample was dipped using a glass cell and was stabilized at 277 K using a temperature controller. ATR-FTIR spectra were recorded in kinetics mode at 2 cm$^{-1}$ resolution, range of 4000–700 cm$^{-1}$, using an FTIR spectrophotometer (Agilent, CA, USA) equipped with a liquid nitrogen-cooled mercury-cadmium- telluride

(MCT) detector (Bio-Rad FTS6000, Agilent). After the 30 sec illumination with a fiber light delivered through a >560-nm cutoff filter, ATR-FTIR spectra were recorded. Difference ATR-FTIR spectra were calculated as spectra *in light* minus spectra *in dark*.

Light-induced structural changes of pure Lumi, Meta I, and Meta II forms of bRho were measured by conventional low-temperature light-induced FTIR difference spectroscopy. A 40–60 μL aliquot of bRho at a concentration of 10 μM was placed on a BaF$_2$ window and dried with gentle aspiration at 293 K. The film was hydrated with 1 μL H$_2$O and FTIR spectroscopy conducted. The sample was placed in a cryostat (OtistatDN2, Oxford Instruments, Abingdon, UK) mounted in the FTIR spectrophotometer (Cary670, Agilent). For the formation of the Lumi, Meta I, and Meta II intermediates, the sample was illuminated with >560-nm light (cutoff filter) from a 1-kW halogen- tungsten lamp for 1 min at 200 K, 240 K, and 277 K, respectively.

## Hydrolysis assay

Briefly, immunoaffinity-purified bRho in DDM in the absence or presence of Nb2 was illuminated with 125 μW 530-nm fiber light for 10 sec; then treated with NaBH4 in isopropanol at different time points to trap any remaining internal Schiff base-adducted chromophore. Concurrently, this treatment also isolates bRho from detergents via alcohol-induced protein precipitation, and the protein pellet is washed with methanol and water, followed by a 24 h proteolytic digestion with pronase. The proteolysis products were analyzed by LC-MS to capture the amount of remaining bound chromophore via detection of the amounts of Nε-retinyl-lysine[42].

## Intrinsic Trp fluorescence (FTrp) quenching assay

The bRho samples with or without Nb2 were diluted to 5 nM concentration by buffer (containing 0.1 M NaCl, 0.2 M NaH$_2$PO$_4$ pH 7.0, 0.05% DDM), followed by incubation for 30 min in the fluorescence spectrophotometer (FP6500, Jasco, Tokyo, Japan), operating at excitation and emission wavelengths of 295 nm and 330 nm, respectively. The molar ratio of Nb2 to bRho was 5:1. Samples were illuminated for 30 sec with a fiber light delivered through a >560-nm cutoff filter before the fluorescence measurements. The relative $t_{1/2}$ values for the decay of bRho* or bRho*/Nb2 were determined by fitting the time-course data to a single exponential curve.

## Immunofluorescence

WT-bOpsin-expressing or P23H-bOpsin-expressing HEK293S cells were seeded at 1,000 cells per well in a 24- well plate and incubated at 37 °C with 5% CO$_2$ overnight. Cells were infected with Nb2-pMXsIG- or Nb16-pMXsIG-viral preparations in the presence of 10 mg mL$^{-1}$ of polybrene for 2 days. Cells were then fixed with 4% paraformaldehyde/PBS at 400 μl per well for 20 min at RT, followed by permeabilization with 0.01% Triton X-100 for 15 min. After blocking with 10% goat serum for 1 h, cells were incubated with anti-Rho 1D4 antibody overnight at 4 °C. Opsin immunostaining was visualized by incubating cells with 5 μg mL$^{-1}$ Alexa Fluor 647-conjugated goat anti-mouse IgG antibody (Invitrogen, Catalog no. A212336) for 1 h. Three washes with PBS were performed between each step of incubation with primary and secondary antibodies. Fluorescent images were captured in 24-bit depth with a Keyence BZ-X800 All-in-One Fluorescence Microscope (Keyence, Osaka, Japan) using 4X Nikon Plan Fluor lens and a BZ-X Cy5 filter (#OP-87766). Imaging settings were as follows: excitation 620/60 nm, emission 700/ 75 nm, dichroic mirror wavelength 660 nm. Ten fields were taken of each well to generate cell images from three biologically independent replicates. Raw images are available in the Source Data file.

## Software

The following software were used for data analysis:

 XDS: v20210323 and v20220220

 BALBES: v1.0.0

Phaser: v2.8.3
Refmac5: v5.8.0352
Coot: v0.9.8
Privateer: vMKIV: 06/02/2021
Molprobity server: v4.5
wwPDB validation pipeline: v2.31.3.

## Reporting summary

Further information on research design is available in the Nature Portfolio Reporting Summary linked to this article.

## Data availability

The data that support the findings of this study are available within the Source Data file included with this manuscript. The crystal structures and their associated diffraction data used in this study are available in the Protein Data Bank (PDB) under accession codes 1L9H (bRho), 4BEL (Bace2), 1U19 (bRho), 5TE3 (Bos taurus opsin), 8FCZ (bRho/Nb2), 8FD1 (bRho*/Nb2), and 8FDO (opsin/Nb2). Biological materials used to support the findings of this manuscript are available from the corresponding author(s) upon request. Source data are provided with this paper.

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

## Acknowledgements

We thank Savhanna Pejoro (UC Irvine) for technical assistance with this project and Eva Beke (VIB) for technical assistance during the Nb discovery. This research was supported in part by grants from the National Eye Institute (R01EY034519 to P.D.K. and K.P.), the Department of Veterans Affairs (I01BX004939 to P.D.K.) and the National Science Foundation (CHE-2107713 to P.D.K.). The authors acknowledge support from NIH grant P30EY034070 and from an unrestricted grant from Research to Prevent Blindness to the Gavin Herbert Eye Institute at the University of California, Irvine. We acknowledge INSTRUCT-ERIC, part of the European Strategy Forum on Research Infrastructures (ESFRI to J.S.), and the Research Foundation Flanders (FWO to J.S.) for their support for the Nanobody Discovery. Use of beamlines 12-1 and 12-2 at the SSRL, SLAC National Accelerator Laboratory, is supported by the U.S. DOE Office of Science under Contract No. DE-AC02-76SF00515. The SSRL Structural Molecular Biology Program is supported by the DOE Office of Biological and Environmental Research, and by the National Institutes of Health (P41GM103393). This work is based upon research conducted at the NE-CAT beamlines, which are funded by the NIH (P30 GM124165). This research used resources of the APS, a U.S. DOE Office of Science User Facility operated by Argonne National Laboratory under contract no. DE-AC02-06CH11357. This research study was also supported in part by the Japanese Ministry of Education, Culture, Sports, Science, and Technology (18K14662 to K.K.) and (21H04969 toH.K.); by a grant-in-Aid for Scientific Research on Innovative Areas "Non-equilibrium-state molecular movies and their applications (Molecular Movies)" from MEXT, Japan (20H05440 to K.K.), and from the Japan Science and Technology Agency (JST), PRESTO (JPMJPR19G4 to K.K.). The contents of this publication do not necessarily represent the official views of any funding agency.

## Author contributions

Conceptualization was provided by A.W., D.S., P.D.K., and K.P.; Nb screening by chromatography and BN-PAGE by D.S. and A.W.; UV-Vis spectroscopy by D.S. and A.W.; Purification of Rho and Nb by D.S. and A.W.; crystallization by D.S. and A.W.; mutational analysis and P23H-bOpsin rescue assay by A.W.; ATR-FTIR and Trp fluorescence-quenching assay by K.W, K.K, H.K.; SPR assay by A.T.; generation and discovery of Nbs by E.P. and J.S.; LC-MS hydrolysis assay by J.D.H.; and crystal structure determination and analysis by P.D.K. The manuscript was written by A.W., D.S., P.D.K., K.P., and all authors commented on the manuscript.

## Competing interests

K.P. is a consultant for Polgenix Inc. and serves on the Scientific Advisory Board at Hyperion Eye Ltd. The remaining authors declare no competing interests.
