## [Peer Review File · Nature Communications]

Structural basis for the allosteric modulation of rhodopsin by nanobody binding to its extracellular domainREVIEWER COMMENTS

Reviewer #1 (Remarks to the Author):

The manuscript by Wu et al reports the development of camelid nanobodies directed against the bovine rhodopsin (bRho). First, the crystal structure of the nanobody Nb2 in complex with the bovine rhodopsin ground state has been solved. The structure of this complex has been validated by site-directed mutagenesis of the three CDRs of Nb2, all three are involved in the interaction with bRho. Using additional biophysical techniques as UV-visible and FITR spectroscopies, the authors demonstrate that Nb2 stabilizes the Meta-I state, and slows down the hydrolysis of the Schiff base of the activated bRho. Both bovine native membranes and bRho in detergent solution were used. Crystal structure of the complex between apo-Rho and Nb2 and comparison with that of apo-Rho alone has revealed that Nb2 prevents conformational changes in the TM5 and TM6. Finally, the authors have shown that Nb2 acts as a molecular chaperone on a genetic mutant of Rho involved in retinitis pigmentosa.

The discovery of nanobodies against rhodopsin is interesting. The characterization of the complex between one nanobody (Nb2) and Rho, and the conformational effect of Nb2 was extensively study with specialized techniques. There is a lot of technical details in the Results section not easy to follow for a non-rhodopsin specialist. It would have been important to verify the chaperone effect of Nb2 on several genetic mutants since many were reported including in the EL2 bound by Nb2. Finally, several important issues should be clarified, as well as some points in writing.

Major points:

- Last sentence in the Abstract “These findings shed light on the role of the extracellular surface in controlling rhodopsin activation” is overstated. It is not clear how the results of this study bring new information on the role of the extracellular part of the rhodopsin on its activation.

- It is surprising to not observe the large conformational change in the TM5 and TM6 in the crystal structure of apo-Rho-Nb2 compared to apo-Rho (Figure 4d) since only small differences in EL2 and EL3 are observed between these two structures. How such small differences in the extracellular part could such a major change in the intracellular part of TM5 and TM6? Could the main differences observed in TM5 and TM6 be due to different conditions of crystallization between the two crystal structures, since the apo-Rho structure used in this study is from a PDB structure previously reported.

- The study focuses on Nb2, and other anti-Rho nanobodies are also used in the different experiments, but the nanobodies used are not always the same in the different experiments, introducing confusion in

the manuscript. For example, the rationale to investigate Nb9 (Figure 3) and other nanobodies (Supp Figure 4) in FTIR is not clear. Why not using Nb16 that does not bind Rho as a negative control in FTIR experiments?

- Analysis of the genetic mutant P23H: Fig.1a should show the expression of the WT and P23H alone (without Nb co-transfected).

- It is unclear why the genetic mutant P23H was chosen since many other Rho mutants from the same class (class II) were reported in the N-term and EL2 (see PMID: 35595348)? It could be important to test additional genetic mutants, at least one in EL2 where Nb2 binds, to see how general the chaperone effect is.

- Nb2 mutants (Figure 1d-f): there is an apparent contradiction for the mutant Nb2 G101A (see Supp Fig. 2a) where Nb2 G101A remains able to co-IP rhodopsin strongly (as much as other mutants shown to not impair strongly Rho binding such as T104A for example). Could you clarify this discrepancy between Figure 1d-f and Supp Fig. 2a?

- It is not clear with which region of Rho residues G99, Y100 and G101 of Nb2 interact with. Is it EL2?

- Affinity determined by surface plasmon resonance (SPR): why the affinity of Nb9 and Nb17, used in FTIR microscopy is not investigated? Is it because they are supposed to have a lower affinity than Nb2 (117 nM)? In addition, do SPR experiments are considered as dark or light conditions?

- Supp Fig 1b-c: Nb9 and Nb17 are not shown even though they are further investigated in FTIR microscopy.

- Supp Fig 1b-c: how to explain the lower shift of the Rho with Nb2 compared to the other nanobodies? In the Figure legend, the authors claim that it may come from the difference of isoelectric points (PI). The PI values should be provided. Another possible explanation could be the lower affinity of Nb2 compared to Nb7, 12 and 22, even though the authors claim that the relative strength of Nb binding to Rho is similar between Nb2 and Nb22. Do the authors could clarify this point?

- Figure 2: Why it is important to study the influence of the temperature and pH?

Minor points:

- Fig. 1g: add molecular weight.
- Page 4: Rho*, "Photoactivated Rho" should be defined earlier (in the Introduction, page 3).
- Page 5: in the second paragraph, it is mentioned "five" mutants but six are listed.
- Figure 2: Meta I is indicated at 478 nm (panel d) and measured at 475 nm (panel f and h). Could you clarify?
- Page 7: could you explain briefly the meaning of "ATR-FTIR"?
- Figure 2d: could you indicate that Meta II activates transducin? Indeed, it is not clear for a non-specialist which state couples to G protein.
- Typographical errors: Fig. 3c "Measurment", "with or with Nb2" and "60 min" (p. 34).

Reviewer #2 (Remarks to the Author):

The manuscript by Wu and coworkers provides interesting insight into the role of the extracellular surface in controlling rhodopsin activation by the use of nanobodies.

SPR was employed to investigate and characterize the interaction of the nanobody Nb2 with bRho. In view of the fact two glycosylation sites seem to be crucial for the interaction of the nanobody as per the co-IP assay that the authors have performed, the authors are strongly encouraged to perform SPR measurements using the two mutants N15Q and N2Q.

Reviewer #3 (Remarks to the Author):

Only the FTIR part was evaluated as the rest of the manuscript is out of my expertise's field.

The difference spectra are difficult to understand as the Y scale is not clear. The FTIR spectra should be always provided and not only the difference.

The intensity of the changes looks very weak and therefore can be just noise error. In Fig 3 the Lumi state seems to have a good signal to noise ratio (black curve) but the Meta-I and Meta-II have poor signal to noise as we can see a lot of variations through the spectra even where there is no absorption band. For example in the meta-II curve, there is a negative peak on the all Amide II band (1600-1500 cm^{-1}) if secondary structure occurs there should be positive and negative peak in this region.

The experiment should be repeated at least 3 times and student test done to confirm that changes are significant.

The FTIR spectra of all Nb alone should also be provided to be able to discuss structural changes (or at least their secondary structures).

The 967 cm^{-1} is expected from the Chromophore, which chromophore is expected? because retinaldehyde will have some band in the yellow region and should be discussed.

Fig S4 in the legend it is explained FTIR difference spectra of bRho/Nb2 in red. I guess it is bRho/Nb# where # is the number of the Nb tested. The difference spectra of some Nb looks to contain only noise like Nb3 or Nb10.

The Lumi and Meta-I spectra are very similar but Meta-I spectra seems just noisier. Replicates of the measurements should be done and the variation mentioned confirmed by statistical test.

Therefore the conclusion from the FTIR need to be confirmed by statistical test and replicates.

In general, references should be provided for the different band attribution (like the hydrogen out of plane band or the β -sheet). Also in the previous paragraph with the UV-vis results, no references are provided for the band attribution.

Reviewer #4 (Remarks to the Author):

This paper presents a detailed characterization and validation of nanobodies targeted to the extracellular domain of bovine rhodopsin (bRho), which function effectively as allosteric modulators. The authors have extensively validated the binding specificity and functional impact of these nanobodies under different experimental conditions using a variety of complementary techniques. The key results include:

- Resolving the crystal structure of the ground state Nb2-Rh complex
- Elucidating the role of EL2 and of N-linked glycans (at N2 and N15), which are critical epitopes for the specificity of the Nbs
- Nb binding stabilizes a photoactivated Rho in an inactive Meta-I state, suppressing proton transfer and retinal hydrolysis
- Improved expression of the misfolding-prone P23H mutant upon Nb co-expression

Significance

The detailed structural and functional characterization of Nb binding to the extracellular loop 2 (EL2) provides insight into the role of EL2 in Rho activation. The highly specific interaction of the Nb with N-linked glycans is another important finding, as it highlights the role of posttranslational modifications in stabilizing native GPCRs. The authors further demonstrate a protective role of extracellular Nb binding recurring a misfolding defect in a disease model of the mutant P23H rhodopsin. This research has important implications for the development of targeted extracellular nanobodies for GPCR stabilization and as potential therapeutic interventions. Overall this manuscript is very well written and is suitable for the readership of Nature Communications.

I recommend publication of this paper, upon addressing some minor comments listed below:

- From Figures S1, S6, Nb7 appears to be a stronger binder and is more protective. Why was Nb2 chosen for the characterization? Nb7 may be a better candidate for the P23H-opsin rescue experiment
- The increased opsin membrane expression and extended cell morphology in Figure 6c is interesting and could use some statistics to compare to 9-cis retinal treatment.
- Page4, Results line 5 should read “inactivated”
- Legend for Co-IP data in Fig1 (g,i) can include a small clarification of how the input/elution panel relates to the gel image
- Fig4: Panels d,e,f: The colour scheme and representation is a little bit chaotic, can be made more colour blind friendly

- Insert reference for the published figure in Figure 4
- Page 8, second line is unclear: “The band at 1636 (+) cm⁻¹ in the spectra of bRho*/Nb2, which is not present in the reference spectra..” .. which reference spectra does this refer too, as the band is seen in bRho* Lumi-I and Meta-I
- Fig 6(e,f) LCMS chromatogram of WT/P23H: what is the peak between 22-24 min which shows up only for the +Nb2?

REVIEWER COMMENTS

Reviewer #1 (Remarks to the Author):

The manuscript by Wu et al reports the development of camelid nanobodies directed against the bovine rhodopsin (bRho). First, the crystal structure of the nanobody Nb2 in complex with the bovine rhodopsin ground state has been solved. The structure of this complex has been validated by site-directed mutagenesis of the three CDRs of Nb2, all three are involved in the interaction with bRho. Using additional biophysical techniques as UV-visible and FITR spectroscopies, the authors demonstrate that Nb2 stabilizes the Meta-I state, and slows down the hydrolysis of the Schiff base of the activated bRho. Both bovine native membranes and bRho in detergent solution were used. Crystal structure of the complex between apo-Rho and Nb2 and comparison with that of apo-Rho alone has revealed that Nb2 prevents conformational changes in the TM5 and TM6. Finally, the authors have shown that Nb2 acts as a molecular chaperone on a genetic mutant of Rho involved in retinitis pigmentosa.

The discovery of nanobodies against rhodopsin is interesting. The characterization of the complex between one nanobody (Nb2) and Rho, and the conformational effect of Nb2 was extensively study with specialized techniques. There is a lot of technical details in the Results section not easy to follow for a non-rhodopsin specialist. It would have been important to verify the chaperone effect of Nb2 on several genetic mutants since many were reported including in the EL2 bound by Nb2. Finally, several important issues should be clarified, as well as some points in writing.

We appreciate the reviewer's summary and helpful critiques of our manuscript. We have revised the Results section with the reviewer's comment in mind about potential difficulties for non-rhodopsin specialists to understand the technical details. The other critiques are addressed below.

Major points:

- Last sentence in the Abstract "These findings shed light on the role of the extracellular surface in controlling rhodopsin activation" is overstated. It is not clear how the results of this study bring new information on the role of the extracellular part of the rhodopsin on its activation.

We thank the reviewer for the comment. Prior mutagenesis studies on residues within extracellular loop 2 (EL2) of bovine rhodopsin (e.g., Glu181, Cys187, Ser186, and Tyr191 located near the chromophore) and other GPCRs have suggested a critical role for EL2 in ligand binding^{1, 2, 3, 4}. Mutations in these regions of EL2 showed negative regulation or constitutive activation of the receptors. However, mutagenesis studies have limitations since they might disrupt the structure of the EL2 or cause unanticipated secondary effects. In other words, these mutagenic experiments cannot confirm whether the specific mutated residues were necessary for the structural integrity of the orthosteric site or for an allosteric transition event upon ligand binding. In particular, it was still not clear how the rigid EL2 in rhodopsin plays a role within the orthosteric pocket where the ligand is covalently attached. The nanobody (Nb2) we have employed has its major site of receptor engagement with EL2. Although reversible nanobody binding to the otherwise native extracellular surface did not disrupt the structural

integrity of EL2 or alter the spectral features of ground-state Rho, it nevertheless produced a dramatic effect on the Rho photoactivation process. Based on the mode of Nb2-Rho interaction (*via* EL2), we can confidently infer that the dynamics of EL2, which are suppressed by Nb2 binding, are critically important for Rho photoactivation from the ground-state to a Lumi-like early intermediate. To address the review's concern, we have modified the final sentence in the abstract to more precisely define the advances made in our study as follows:

"Our data show the power of nanobodies to modulate the photoactivation of rhodopsin and potentially to serve as therapeutic agents for rescue of disease-associated rhodopsin misfolding."

References

1. Ahuja S., *et al.* Helix movement is coupled to displacement of the second extracellular loop in rhodopsin activation. *Nat. Struct. Mol. Biol.* **16**, 168-175 (2009).
2. Avlani V. A., Gregory K. J., Morton C. J., Parker M. W., Sexton P. M., Christopoulos A. Critical role for the second extracellular loop in the binding of both orthosteric and allosteric G protein-coupled receptor ligands. *J. Biol. Chem.* **282**, 25677-25686 (2007).
3. Klco J. M., Wiegand C. B., Narzinski K., Baranski T. J. Essential role for the second extracellular loop in C5a receptor activation. *Nat. Struct. Mol. Biol.* **12**, 320-326 (2005).
4. Shi L., Javitch J. A. The second extracellular loop of the dopamine D2 receptor lines the binding-site crevice. *Proc. Natl. Acad. Sci. U. S. A.* **101**, 440-445 (2004).

- It is surprising to not observe the large conformational change in the TM5 and TM6 in the crystal structure of apo-Rho-Nb2 compared to apo-Rho (Figure 4d) since only small differences in EL2 and EL3 are observed between these two structures. How such small differences in the extracellular part could such a major change in the intracellular part of TM5 and TM6? Could the main differences observed in TM5 and TM6 be due to different conditions of crystallization between the two crystal structures, since the apo-Rho structure used in this study is from a PDB structure previously reported.

Although we were unable to crystallize bovine opsin by itself in the same conditions as the opsin/Nb2 complex, our structural results are fully supported by biochemical data showing that Nbs can shift the equilibrium of photoactivated rhodopsin from a Meta II to a Meta I-like state *in solution*. Note that the opsin/Nb2 crystals were obtained from opsin/Nb2 complexes in solution, not by bleaching preformed bRho/Nb2 crystals in the presence of hydroxylamine.

We also don't believe that the observed difference is attributable to the choice of precipitant (PEG 600). Although most of the published crystal structures of bovine opsin and photoactivated rhodopsin have been obtained using ammonium sulfate as precipitant, one crystal structure of activated rhodopsin was obtained using PEG 4000 (PDB 6FUF, a constitutively active mutant of rhodopsin co-crystallized with an engineered mini-Go protein)⁵. When compared to bovine opsin and other activated forms of rhodopsin, 6FUF folding is essentially identical in terms of the large intracellular helix movement⁵.

Another significant difference in the crystallization conditions between our opsin/Nb2 crystal and previous crystals of photoactivated rhodopsin (Rho*) and opsin is the pH. Our crystals grew

optimally at pH 7.8 - 8.2 (close to physiological pH), while all the published crystals of Rho* and opsin grew at pH 4.5 - 6.0. Therefore, we are confident that the absence of large movements of TM5 and TM6 in our crystal structure of Nb2 in complex with opsin is not due to a crystallization artifact.

Reference

5. Tsai C. J., *et al.* Crystal structure of rhodopsin in complex with a mini-Go sheds light on the principles of G protein selectivity. *Sci. Adv.* **4**, eaat7052 (2018).

- The study focuses on Nb2, and other anti-Rho nanobodies are also used in the different experiments, but the nanobodies used are not always the same in the different experiments, introducing confusion in the manuscript. For example, the rationale to investigate Nb9 (Figure 3) and other nanobodies (Supp Figure 4) in FTIR is not clear. Why not using Nb16 that does not bind Rho as a negative control in FTIR experiments?

We thank the reviewer for the comment and have modified Figure 3 by adding ATR-FTIR spectra for Nb16. Supplementary Figure 4 shows the ATR-FTIR spectra of Nbs including Nb2, Nb9 and Nb16. Our rationale for investigating the other Nbs relates to their differential stabilization activity. It is interesting that subsets of Rho-binding Nbs (Nb9 and Nb17) stabilized a Meta-I-like conformation in photoactivated bRho* in membranes, whereas most of the other Nbs stabilized a Lumi-like conformation. Consequently, we believe it is worthwhile to show the ATR-FTIR spectra of Nb9 in Figure 3. We agree with the reviewer that showing Nb2, Nb9, and Nb16 side-by-side in Figure 3 as examples for stabilizing Lumi-like or Meta I-like conformations, or non-binding, respectively, illustrates the power of our Nb approach in trapping distinct states.

- Analysis of the genetic mutant P23H: Fig.1a should show the expression of the WT and P23H alone (without Nb co-transfected).

We have updated Figure 6a-b with a control (not co-transfected).

- It is unclear why the genetic mutant P23H was chosen since many other Rho mutants from the same class (class II) were reported in the N-term and EL2 (see PMID: 35595348)? It could be important to test additional genetic mutants, at least one in EL2 where Nb2 binds, to see how general the chaperone effect is.

There are several reasons why the P23H mutant was chosen for this study over other class II Rho mutants. First, the P23H mutation in the *RHO* gene is the most common variant associated with adRP in North America, while more than 180 mutations in *RHO* are involved in adRP. The underlying pathologic mechanism of P23H-Rho is well characterized - disrupted proteostasis *via* unfolded protein response (UPR). There are several ongoing clinical investigations in the pharmaceutical industry for mitigating the deleterious effect of the P23H mutation for adRP patients. Thus, the P23H-Rho mutation is clinically most relevant to study regarding Rho-adRP.

We cannot agree more with the reviewer regarding the importance of testing additional Rho mutants. In this study, we have shown the potential therapeutic properties of Nb2 for Rho-adRP using bovine WT and P23H-mutant opsin cells. However, the chaperone effect of Nb2 is limited to bovine rod opsin because our nanobodies were raised against bovine rod outer segments. As such, human Rho is not recognized by Nb2. Therefore, the testing of Nb2 activity towards different Rho mutants is not feasible and beyond the scope of the current study. For our subsequent studies, we are planning to develop human Rho-binding Nbs against the same epitope, which then can be investigated for the broad spectrum of Rho gene mutations.

Additionally, we expect that Nb2 might not bind to the Rho mutants with mutations in EL2 or near Asn2 or Asn15, as Nb2 recognizes the conformational epitopes consisting of both regions (as shown in Figure 1d, g). The chaperone effects of Nb2 probably will be conserved for wild-type Rho and the broad spectrum of Rho mutants, unless there is a disruption of N-glycosylation levels or EL2 structure.

- Nb2 mutants (Figure 1d-f): there is an apparent contradiction for the mutant Nb2 G101A (see Supp Fig. 2a) where Nb2 G101A remains able to co-IP rhodopsin strongly (as much as other

mutants shown to not impair strongly Rho binding such as T104A for example). Could you clarify this discrepancy between Figure 1d-f and Supp Fig. 2a?

We thank the reviewer for the comment. We have updated Supplemental Figure 2a with more representative data. There were experimental variations among independent assays ($n > 4$), so the results shown in Figure 1d were expressed as the relative amounts of bound Rho (~35 kDa apparent MW), normalized by the eluted amount of the mutant Nbs. This normalization was necessary to perform proper statistical analysis.

In these updated co-IP results (shown above), the G101A mutant exhibited diminished binding to Rho compared to T102A – E106A, which is the same outcome as shown in Figure 1d. G101 still seems critical for Nb2 binding to bRho by participating in recognition of the epitope. This issue is discussed more fully in our response to the next question.

- It is not clear with which region of Rho residues G99, Y100 and G101 of Nb2 interact with. Is it EL2?

We thank the reviewer for pointing out this ambiguity. The manuscript text has been clarified as follows:

“We observed that Phe²⁷, Thr²⁸, Lys³¹, and Tyr³² in CDR1 of Nb2 interact with Pro¹⁹⁴, His¹⁹⁵, Glu¹⁹⁶, and Glu¹⁹⁷ of EL2 in bRho via electrostatic and van der Waals interactions. Glu¹⁹⁷ of EL2 and Glu²⁰¹ located at the top of TM5 also formed electrostatic interactions with Arg⁹⁸ of CDR3 and Lys³¹ of CDR1. In addition to these interactions with EL2 and TM5 of bRho, Gly⁹⁹, Tyr¹⁰⁰, Gly¹⁰¹ and Met¹⁰³ of Nb2-CDR3 and Asp⁶² and Trp⁴⁷, which are outside the classical CDR regions of Nb2, interact with the N-terminus of bRho (Fig. 1c and 1f). Tyr100 of Nb2 also interacts with Asn279 and Gly280 in EL3 of bRho.”

- Affinity determined by surface plasmon resonance (SPR): why the affinity of Nb9 and Nb17, used in FTIR microcopy is not investigated? Is it because they are supposed to have a lower
- Supp Fig 1b-c: Nb9 and Nb17 are not shown even though they are further investigated in FTIR microscopy.

We performed BN-PAGE for all nanobodies and only Nb2, Nb7, Nb12, and Nb22 produced dose-dependent gel shifts of bRho among 17 strong binders that were grouped together by co-

IP results. The rest of them did not produce a dose-dependent gel shift. Here, we show the BN-PAGE results of Nb9 and Nb17 (4 µg each) with native bRho (1 µg) in A in the Figure below. Nb12 was used as a positive control for BN-PAGE. Neither Nb9 nor Nb17 produced gel shifts of bRho (A). Nb17 also failed to produce a dose-dependent gel shift of native bRho (1 µg) with two-fold increasing amounts of Nbs (0.125 - 8 µg) (B). All BN-PAGE assays were performed in the dark.

- Supp Fig 1b-c: how to explain the lower shift of the Rho with Nb2 compared to the other nanobodies? In the Figure legend, the authors claim that it may come from the difference of isoelectric points (PI). The PI values should be provided. Another possible explanation could be the lower affinity of Nb2 compared to Nb7, 12 and 22, even though the authors claim that the relative strength of Nb binding to Rho is similar between Nb2 and Nb22. Do the authors could clarify this point?

We thank the reviewer for the comment. BN-PAGE is a tool to study protein-protein interaction based on the size, shape, and charge under native conditions near neutral pH. The sizes of all 4 nanobodies are similar so any differences in mobility on the gel are attributable to differences in conformation and/or net charge of the respective Nb-Rho complexes. The NativePAGE gel system uses Coomassie G-250 as a charge-shift molecule, conferring a net negative charge and causing the protein complexes to migrate towards the Anode. The G-250 is present in the cathode buffer.

Nb7, Nb12, and Nb22 have slightly more basic isoelectric points compared to Nb2, so the net negative charge of their G-250-bound protein complexes is diminished, thereby slowing their migration toward the anode (larger gel shifts). This consideration explains why Nb2 has the smallest gel shift (migrating faster), compared to Nb7, Nb12 and Nb22. Nb22 shows the largest gel shift compared to the rest, due to it having the most basic isoelectric point and a larger protein size. This analysis has been included in the revised manuscript as follows:

“The electrophoretic mobilities of the bRho/Nb complexes relative to bRho alone were found to correlate with the isoelectric points (pI) of the Nbs; the pIs of Nb2, Nb7, Nb12 and Nb22 are 6.64, 7.18, 7.18, and 8.01, respectively.”

	Protein size (kDa)	Isoelectric point (pI)
Nb2	13.83	6.64
Nb7	13.89	7.18
Nb12	13.92	7.18
Nb22	14.77	8.01

Differences in affinity are not well correlated with the gel shift differences. Therefore, conformational or affinity differences of the Nb-Rho complexes are not likely to be the main reasons for the observed differences in gel shift.

- Figure 2: Why it is important to study the influence of the temperature and pH?

UV-visible spectroscopy was used to monitor conformational changes under various conditions upon light activation of rhodopsin. Prior reports showed that temperature and pH affect the stability of photoproducts, including Lumi and Meta I⁶. Since Nb binding itself affects the stability of the intermediate conformations of Rho, it was important to investigate the effects of environmental factors on the shifts in equilibrium of the Nb-bRho complexes, with or without photoactivation. Based on these results, ATR-FTIR experiments for Nb-Rho complexes were performed at room temperature. In Figure 2, we also monitored the UV-Vis spectra of the Nb-Rho complexes for an extended period time for spectral changes and discovered that the photoactivated Nb-Rho complexes maintained stability over the time course. The crystals of bRho-Nb₂, bRho*-Nb₂ and apo-opsin-Nb₂ were grown under different combinations of conditions, including 4 °C or room temperature, and with or without light in a time span of 2-3 weeks before harvest. Thus, the stability of Nb₂-bound intermediate photoproducts under various conditions provided insights for the experimental designs for SPR, ATR-FTIR, and crystallization of bRho* in the current study.

Reference

6. Brown M. F. UV-visible and infrared methods for investigating lipid-rhodopsin membrane interactions. *Methods Mol. Biol.* **914**, 127-153 (2012).

Minor points:

- Fig. 1g: add molecular weight.

We have updated Figure 1g and 1i accordingly.

- Page 4: Rho*, "Photoactivated Rho" should be defined earlier (in the Introduction, page 3).

We have updated the manuscript accordingly in the first paragraph of the Introduction.

- Page 5: in the second paragraph, it is mentioned "five" mutants but six are listed.

We thank the reviewer for catching this mistake and have corrected the manuscript.

- Figure 2: Meta I is indicated at 478 nm (panel d) and measured at 475 nm (panel f and h). Could you clarify?

We thank the reviewer for pointing out this inconsistency. In the literature⁷, the Meta-I absorbance maximum is reported to be 478-480 nm, so the graphs in Figures 2f, 2g, 3d, 5c, S4e and S6b have been updated to 478 nm.

- Page 7: could you explain briefly the meaning of “ATR-FTIR”?

We thank the reviewer for the comment and have properly defined ‘ATR-FTIR’ at its first use in the manuscript. Attenuated Total Reflection (ATR)-Fourier Transform Infrared (FTIR) spectroscopy measures light-induced FTIR difference spectra of Rho *via* a reflected IR beam in contact with Nb-Rho samples. The difference spectra of the after-minus-before illumination is measured, capturing short-lived intermediates of Rho. The diagram shows the experimental set up to measure the ATR-FTIR.

- Figure 2d: could you indicate that Meta II activates transducin? Indeed, it is not clear for a non-specialist which state couples to G protein.

We have updated Fig. 2d according to the reviewer’s suggestion.

- Typographical errors: Fig. 3c “Measurment”, “with or with Nb2” and “60 min” (p. 34).

We thank the reviewer for catching these typos and have corrected the manuscript in Figure 3c and Figure legend 6d-g.

Reviewer #2 (Remarks to the Author):

The manuscript by Wu and coworkers provides interesting insight into the role of the extracellular surface in controlling rhodopsin activation by the use of nanobodies. SPR was employed to investigate and characterize the interaction of the nanobody Nb2 with bRho. In view of the fact two glycosylation sites seem to be crucial for the interaction of the nanobody as per the co-IP assay that the authors have performed, the authors are strongly encouraged to perform SPR measurements using the two mutants N15Q and N2Q.

We thank the reviewer for the helpful and supportive comments. Our co-IP with N2Q and N15Q bRho resulted in weak to no binding of Nb2. Thus, we expect that reliable SPR data will not be possible to record for these mutants. In lieu of SPR experiments, we carried out BN-PAGE on the two mutants, which can easily be performed in the dark, thereby maximizing the possibility of observing a binding signal. The results are shown below. Nb2 produced a gel shift only for WT bRho but not for N2Q or N15Q Rho, likely due to loss of binding affinity for Rho.

Reviewer #3 (Remarks to the Author)

Only the FTIR part was evaluated as the rest of the manuscript is out of my expertise's field.

- The difference spectra are difficult to understand as the Y scale is not clear. The FTIR spectra should be always provided and not only the difference.

Thank you for the reviewer's comment. We have updated Figure 3 (shown below) which includes the Y scale bar. Additionally, we added scale bars for the ATR-FTIR spectra in Supplementary Fig 6.

In the figure below, we also show the FTIR absorbance spectra of bovine ROS in the dark, in light, and the light-minus-dark difference for comparison (upper panel). The lower panel is a magnification of the difference spectra in the 900-1800 cm^{-1} region (blue box). The IR spectra show vibrations from the chromophore, protein, lipid, and water molecules. The light-induced difference spectrum (lower panel) allows us to analyze subtle conformational changes by removing the spectral features common to both the dark and light-activated states.

- The intensity of the changes looks very weak and therefore can be just noise error. In Fig 3 the Lumi state seems to have a good signal to noise ratio (black curve) but the Meta-I and Meta-II have poor signal to noise as we can see a lot of variations through the spectra even where there is no absorption band. For example in the meta-II curve, there is a negative peak on the all Amide II band (1600-1500 cm⁻¹) if secondary structure occurs there should be positive and negative peak in this region.

Thank you for the reviewer's comment. In Figure 3, black curves indicate reference spectra of Rho only in Lumi, Meta I and Meta II states. As the reviewer pointed out, although the typical Amide-II band (N-H bending vibration and C-N stretching vibration) appears at 1600-1500 cm⁻¹,

this frequency region is overlapped with strong C=C stretching vibrational bands of the retinal chromophore. In the case of Meta-II, the retinal Schiff base is deprotonated; consequently, vibration bands originating from retinal such as C-C stretch and C=C stretch become insensitive to infrared absorption. Therefore, the positive band corresponding to the Meta-II state is diminished, and the negative band corresponding to the dark state is enhanced. As a result, in the case of the Meta-II minus Rho spectra, the Amide-II band would be cancelled by enhancement of the negative C=C stretching vibration in the same frequency region.

In addition, we also compared the light-induced FTIR difference spectrum with the baseline difference spectrum which was obtained by calculation of dark minus dark spectra before illuminations, as shown below. As is clearly seen, the peak intensity observed in the light-induced FTIR difference spectra has a good signal-to-noise ratio as compared to the calculated baseline difference spectrum. Prior reports also have shown spectral features like those of the pure Lumi, Meta-I and Meta-II spectra recorded in our current study^{8, 9, 10}.

References

7. Wald G. Molecular basis of visual excitation. *Science* **162**, 230-239 (1968).
8. Ye S., *et al.* Tracking G-protein-coupled receptor activation using genetically encoded infrared probes. *Nature* **464**, 1386-1389 (2010).
9. Furutani Y., Kandori H., Shichida Y. Structural changes in lumirhodopsin and metarhodopsin I studied by their photoreactions at 77 K. *Biochemistry* **42**, 8494-8500 (2003).
10. Kazmin R., *et al.* The Activation Pathway of Human Rhodopsin in Comparison to Bovine Rhodopsin. *J. Biol. Chem.* **290**, 20117-20127 (2015).

- The experiment should be repeated at least 3 times and student test done to confirm that changes are significant.

We thank the reviewer for this comment. In Figure 3 and Figure S4, the individual experiments for all Nb-Rho complexes were independently performed at least 3 times and averaged. By “independently” we mean that fresh ROS and Nbs were used for each measurement because the ROS was bleached after illumination. In the 12-panel figure below, we show the individual (black) and average (red) spectra of all Nb-Rho samples: n=3 (Nb1~16, and 22); n=5 (Nb17); or n=6 (Nb28). It will be evident to the reviewer that the precision of the individual measurements is exceptionally high for the Nb samples of main interest in this study, resulting in small uncertainties for the major peaks of interest. The Student’s t-test is not suitable for comparing two spectra with various specific fingerprint peaks.

- The FTIR spectra of all Nb alone should also be provided to be able to discuss structural changes (or at least their secondary structures).

We thank the reviewer for the comment. The spectra of all Nbs alone are nearly identical so here we show the representative absolute absorbance spectra of Nb2 alone for both dry-layer and hydrated with buffer. The positive 1635 or 1637 cm^{-1} bands are characteristic for the amide-I band of a β -sheet structure, which is reasonable for the secondary structure of Nb. Because Nbs are structurally rigid overall, we expect that specific structural changes upon binding to opsin proteins will not be readily distinguished by the ATR-FTIR method. Also, we don't expect any change in the Nbs alone in the dark or under light conditions.

A comparison of the Nbs in the three crystal structures reported here as well as the Nb used for initial molecular replacement shows very similar structures with small RMSDs, which further argues against Nb structural changes being responsible for features we observe in the ATR-FTIR difference spectra.

- The 967 cm^{-1} is expected from the Chromophore, which chromophore is expected ? because retinaldehyde will have some band in the yellow region and should be discussed.

Thank you to the reviewer for the comment. As prior reports showed^{8, 11, 12}, the positive HOOP bands at 947 cm⁻¹ and 951 cm⁻¹ are seen in Lumi and Meta-I, respectively but not in Meta-II Rho due to deprotonation of the Schiff base. The negative HOOP band at 967 cm⁻¹ originated from a HC11=C12H hydrogen-out-plane-mode in native Rho. Retinal bands at 1238 (-) cm⁻¹ are fingerprint bands (C-C stretching) shown for native Rho; while positive 1206 cm⁻¹, 1198 cm⁻¹, and 1184 cm⁻¹ bands are apparent for Lumi and Meta-I, but not for Meta-II. However, retinal vibrations in the 1500-1600 cm⁻¹ region (C=C stretching) are obstructed by overlapping vibrations from the protein backbone and side chains, including the amide II bands, in photoactivated Rho. Thus, specific band assignment to the retinal chromophore is not available.

This clarification has been added to the revised manuscript as follows:

“Vibrations from the chromophore appear in the 1300-900 cm⁻¹ region. The characteristic positive hydrogen-out-of-plane (HOOP) bands for Lumi and Meta-1 appear at 947 (+) cm⁻¹ and 951(+) cm⁻¹, respectively, but not in Meta II Rho due to deprotonation of the Schiff base. The negative bands at 967 (-) cm⁻¹ and 1238 (-) cm⁻¹ in Nb2 and Nb9-bound bRho originate from a HC11=C12H HOOP mode and C-C stretching, respectively, in native Rho while positive 1206 cm⁻¹, 1198 cm⁻¹, and 1184 cm⁻¹ bands are seen for Lumi and Meta I, but not Meta II.”*

References

8. Ye S., *et al.* Tracking G-protein-coupled receptor activation using genetically encoded infrared probes. *Nature* **464**, 1386-1389 (2010).

11. Kukura P., McCamant D. W., Yoon S., Wandschneider D. B., Mathies R. A. Structural observation of the primary isomerization in vision with femtosecond-stimulated Raman. *Science* **310**, 1006-1009 (2005).

12. Furutani Y., Shichida Y., Kandori H. Structural changes of water molecules during the photoactivation processes in bovine rhodopsin. *Biochemistry* **42**, 9619-9625 (2003).

- Fig S4 in the legend it is explained FTIR difference spectra of bRho/Nb2 in red. I guess it is bRho/Nb# where # is the number of the Nb tested. The difference spectra of some Nb looks to contain only noise like Nb3 or Nb10.

Thank you to the reviewer for the typo. As the reviewer mentioned, FITR difference spectra of bRho/Nb complexes are labeled in red in Figure S4. The left panel shows only pure Lumi in the black curves; the right panel shows only the Meta-I reference spectrum in the black curve. Thus, the spectra of the bRho-Nb3 and bRho-Nb10 complexes (red curves) are shown with Lumi or Meta-I reference spectra (black curves) in this figure, to show that the spectra of bRho-Nb3 and bRho-Nb10 are different from the two photoproducts. We have updated the figure legend for clarification, according to the reviewer's comment.

- The Lumi and Meta-I spectra are very similar but Meta-I spectra seems just noisier. Replicates of the measurements should be done and the variation mentioned confirmed by statistical test. Therefore the conclusion from the FTIR need to be confirmed by statistical test and replicates.

As mentioned above, the peak intensity observed in the light-induced FTIR difference spectra has an excellent signal-to-noise ratio as compared to baseline spectra. Consequently, even the small differences we observe in the peaks of interest are reliably well above the noise level of the recordings, which were performed at least three times independently. Our Lumi and Meta-I reference spectra resemble previously reported data^{8, 10}. As Ye and colleagues reported in a 2010 Nature paper, the Lumi and Meta-I spectra are quite similar, compared to Meta-II spectra. Our Nb-bound Lumi-like and Meta-I-like spectra show diminished vibrations that could be due to structural rigidity caused by Nb binding to the extracellular surface producing a ground-state-like structure as we see *in crystallo*. As discussed above, the Student's t-test is not suitable for comparing two spectra primarily distinguishable by the presence of specific fingerprint peaks.

The fingerprint band at 1636 cm⁻¹ (+) in the spectra of bRho*-Nb2 is a Lumi-specific band that exists as a baseline 1635 (+)/1644 (+) doublet in Meta-I. According to prior reports, 1656 (-) bands (C=O stretch, Amide-I) appear in both Lumi and Meta-I but not in Batho, suggesting that Nb-Rho complexes are in the Lumi or Meta-I states, but not Batho. The 1664 (-) band in Batho gradually shrinks into the baseline in the Lumi state, and emerges as a characteristic 1664 (+) fingerprint band during the transition to Meta-I.

References

8. Ye S., *et al.* Tracking G-protein-coupled receptor activation using genetically encoded infrared probes. *Nature* **464**, 1386-1389 (2010).

10. Kazmin R., *et al.* The Activation Pathway of Human Rhodopsin in Comparison to Bovine Rhodopsin. *J. Biol. Chem.* **290**, 20117-20127 (2015).

- In general, references should be provided for the different band attribution (like the hydrogen out of plane band or the b-sheet). Also in the previous paragraph with the UV-vis results, no references are provided for the band attribution.

Thank you to the reviewer for this comment. We have updated the manuscript by adding the following references for UV-Vis spectroscopy⁷ and ATR-FTIR^{8, 9, 10, 12, 13, 14, 15}

References

7. Wald G. Molecular basis of visual excitation. *Science* **162**, 230-239 (1968).

8. Ye S., *et al.* Tracking G-protein-coupled receptor activation using genetically encoded infrared probes. *Nature* **464**, 1386-1389 (2010).

9. Furutani Y., Kandori H., Shichida Y. Structural changes in lumirhodopsin and metarhodopsin I studied by their photoreactions at 77 K. *Biochemistry* **42**, 8494-8500 (2003).

10. Kazmin R., *et al.* The Activation Pathway of Human Rhodopsin in Comparison to Bovine Rhodopsin. *J. Biol. Chem.* **290**, 20117-20127 (2015).

12. Furutani Y., Shichida Y., Kandori H. Structural changes of water molecules during the photoactivation processes in bovine rhodopsin. *Biochemistry* **42**, 9619-9625 (2003).
13. Fahmy K., Jager F., Beck M., Zvyaga T. A., Sakmar T. P., Siebert F. Protonation states of membrane-embedded carboxylic acid groups in rhodopsin and metarhodopsin II: a Fourier-transform infrared spectroscopy study of site-directed mutants. *Proc. Natl. Acad. Sci. U. S. A.* **90**, 10206-10210 (1993).
14. Jager F., Fahmy K., Sakmar T. P., Siebert F. Identification of glutamic acid 113 as the Schiff base proton acceptor in the metarhodopsin II photointermediate of rhodopsin. *Biochemistry* **33**, 10878-10882 (1994).
15. Rath P., DeCaluwe L. L., Bovee-Geurts P. H., DeGrip W. J., Rothschild K. J. Fourier transform infrared difference spectroscopy of rhodopsin mutants: light activation of rhodopsin causes hydrogen-bonding change in residue aspartic acid-83 during meta II formation. *Biochemistry* **32**, 10277-10282 (1993).

Reviewer #4 (Remarks to the Author):

This paper presents a detailed characterization and validation of nanobodies targeted to the extracellular domain of bovine rhodopsin (bRho), which function effectively as allosteric modulators. The authors have extensively validated the binding specificity and functional impact of these nanobodies under different experimental conditions using a variety of complementary techniques. The key results include:

- Resolving the crystal structure of the ground state Nb2-Rh complex
- Elucidating the role of EL2 and of N-linked glycans (at N2 and N15), which are critical epitopes for the specificity of the Nbs
- Nb binding stabilizes a photoactivated Rho in an inactive Meta-I state, suppressing proton transfer and retinal hydrolysis
- Improved expression of the misfolding-prone P23H mutant upon Nb co-expression

Significance

The detailed structural and functional characterization of Nb binding to the extracellular loop 2 (EL2) provides insight into the role of EL2 in Rho activation. The highly specific interaction of the Nb with N-linked glycans is another important finding, as it highlights the role of posttranslational modifications in stabilizing native GPCRs. The authors further demonstrate a protective role of extracellular Nb binding recurring a misfolding defect in a disease model of the mutant P23H rhodopsin. This research has important implications for the development of targeted extracellular nanobodies for GPCR stabilization and as potential therapeutic interventions. Overall this manuscript is very well written and is suitable for the readership of Nature Communications.

We greatly appreciate the reviewer's summary of our key findings and the significance of our study. We thank the reviewer for their supportive comments.

I recommend publication of this paper, upon addressing some minor comments listed below:

- From Figures S1, S6, Nb7 appears to be a stronger binder and is more protective. Why was Nb2 chosen for the characterization? Nb7 may be a better candidate for the P23H-opsin rescue experiment

We thank the reviewer for this comment. We have chosen Nb2 for detailed characterization throughout our study because Nb2 was the only nanobody that was amenable to crystal structure analysis. Certainly, Nb7 is the strongest binder among all Nbs, and our preliminary data showed that Nb7 also had the chaperone effect on WT and P23H-opsin expressing cells, at a similar level to Nb2 treatment. Both Nb2 and Nb7 are bovine-specific, and our ongoing effort is to develop human/mouse opsin-binding nanobodies for the rescue of mouse and human P23H-mutant opsins.

- The increased opsin membrane expression and extended cell morphology in Figure 6c is interesting and could use some statistics to compare to 9-cis retinal treatment.

We agree with the reviewer that the extended cell morphology by Nb2 treatment is very interesting. For 9-cis-retinal treated cells, the rounder shape is likely due to cytotoxic effects, as we have observed that HEK293S cells show minor signs of cell toxicity upon overnight treatment with 2.5 μ M of 9-cis-retinal. On the other hand, we are trying to be cautious not to overinterpret the apparent stretched-cell morphology of Nb2-expressing cells because it could be due simply to extensive immunofluorescent staining using the strong-binding 1D4 antibody at the plasma membrane. We have clarified the possible reasons for this finding in the revised manuscript.

“Such findings suggest that localization of WT or P23H-bOpsins to the plasma membrane likely occurred to a greater extent as a result of increased 1D4 immunostaining at the membrane with co-expression of Nb2. The rounder morphology of 9-cis-retinal-treated cells may relate to non-specific oxidative stress caused by the retinaldehyde treatment.”

- Page4, Results line 5 should read “inactivated”

We have replaced this typo with “ground-state”, which more accurately describes the form of rhodopsin tested in this experiment.

- Legend for Co-IP data in Fig1 (g,i) can include a small clarification of how the input/elution panel relates to the gel image

We thank the reviewer for this comment and have added the following clarification to the figure legend.

“Equal amounts of Rho samples (Input) were pulled down with His-tagged Nbs (IMAC) and eluted with an excess amount of imidazole.”

- Fig4: Panels d,e,f: The colour scheme and representation is a little bit chaotic, can be made more colour blind friendly

We thank the reviewer for the feedback and have modified the color scheme using guidance from the ColorBrewer2.0 website.

• Insert reference for the published figure in Figure 4

We have inserted the following reference¹⁶.

16. Gulati S., *et al.* Photocyclic behavior of rhodopsin induced by an atypical isomerization mechanism. *Proc. Natl. Acad. Sci. U. S. A.* **114**, E2608-E2615 (2017).

• Page 8, second line is unclear: “The band at 1636 (+) cm^{-1} in the spectra of bRho*/Nb2, which is not present in the reference spectra...” which reference spectra does this refer too, as the band is seen in bRho* Lumi-I and Meta-I

We thank the reviewer for this question. Kazmin *et al.* and other research groups reported Lumi and Meta-I spectra of bovine Rho that resemble our reference spectra ^{8, 10, 12}. The band at 1636 (+) cm^{-1} (C=O stretch, Amide I) in the spectra of bRho*-Nb2 is a Lumi-specific fingerprint peak which exists as a baseline 1636 (+)/1643 (+) cm^{-1} doublet in Meta-I, in alignment with the results from Kazmin *et al.*

We have updated the manuscript to address this issue as follows:

“The band at 1636 (+) cm^{-1} (C=O stretch, Amide I) in the spectra of bRho-Nb2 is a Lumi-specific band, which exists as a baseline 1636(+)/1643(+) cm^{-1} doublet in bRho*-Nb9 and Meta-I reference spectra.”*

Reference

8. Ye S., *et al.* Tracking G-protein-coupled receptor activation using genetically encoded infrared probes. *Nature* **464**, 1386-1389 (2010).

10. Kazmin R., *et al.* The Activation Pathway of Human Rhodopsin in Comparison to Bovine Rhodopsin. *J. Biol. Chem.* **290**, 20117-20127 (2015).

12. Furutani Y., Shichida Y., Kandori H. Structural changes of water molecules during the photoactivation processes in bovine rhodopsin. *Biochemistry* **42**, 9619-9625 (2003).

• Fig 6(e,f) LCMS chromatogram of WT/P23H: what is the peak between 22-24 min which shows up only for the +Nb2?

The peak between 22-24 min was identified as N^ε-9-*cis*-retinyl-Lys, possessing a characteristic UV-Vis absorption spectrum with a λ_{max} of 325 nm, as shown below. Prior reports suggested the formation of isorhodopsin from early rhodopsin photointermediates upon illumination ^{17, 18, 19}. Nb2 stabilization of early intermediates, and the subsequent delayed retinal hydrolysis, probably is the cause of the observed increase of isorhodopsin compared to Rho-only controls.

We have clarified this point in the legend to Figs. 5 and 6 as follows:

“The peak between the 11-cis and all-trans isomer peaks corresponds to N ϵ -9-cis-retinyl-Lys ($\lambda_{\max} = 325$ nm), a byproduct of photoisomerization of early protonated all-trans retinylidene photointermediates.”

We have also updated Fig S6 to include all the three UV-Vis spectra shown above for 9-cis, 11-cis, and all-trans isomers of N ϵ -retinyl-Lys, as well as the MS spectrum for the N ϵ -9-cis-retinyl-Lys peak.

References

17. Catt M., Ernst W., Kemp C. M. The products of photoreversing rhodopsin bleaching by microsecond flashes in the isolated vertebrate retina. *Vision Res.* **23**, 971-982 (1983).
18. Eder D. J., Williams T. P. A method of isorhodopsin analysis and the photoreversal of rhodopsin intermediates. *Am. J. Optom. Arch. Am. Acad. Optom.* **50**, 765-776 (1973).
19. Reuter T. Formation of Isorhodopsin in the Frog's Eye during Continuous Illumination. *Nature* **204**, 784-785 (1964).

REVIEWERS' COMMENTS

Reviewer #1 (Remarks to the Author):

I am satisfied with the authors' responses to my comments, all of which have been addressed.

Reviewer #2 (Remarks to the Author):

The authors have properly addressed the question raised.

Reviewer #5 (Remarks to the Author):

I reviewed the X-ray crystallographic work only. The authors report crystal structures of the complexes between Nb2 and bRho, bRho*, and opsin. These complexes reveal the binding interface between Nb2 and the receptor, and suggest the structural basis for the allosteric modulation of the receptor by Nb2. Even though the resolution of the structures is not very high (3.70, 4.25, and 3.71 Å), they are important for the present study and represent a substantial advancement in the field.

The X-ray crystallographic work appears solid. The chosen resolution cut-offs are supported by the CC1/2 values > 20%, 100% completeness, and good data redundancy (~10) in the highest resolution shell. Only the authors can assess whether the inclusion of reflections to the reported resolution limits is beneficial for the quality of the electron density despite other indicators (e.g., $I / \sigma I$) being somewhat outside of the optimal range.

More than statistical indicators, the quality of the electron density matters. The authors discuss interactions between some of the GPCR and Nb2 residues in detail for the 3.70-Å complex. Given the not-so-high resolution, I suggest that the authors show in the Supplement the electron density (2Fo-Fc map, typically contoured at 1.0 σ) for the GPCR and Nb2 regions discussed in detail. I think that showing the quality of the electron density map is currently common practice in GPCR structure papers. It is nonetheless reassuring that according to the PDB Validation Report of the 3.70-Å complex, most of the discussed residues have a good fit to the electron density, except D56 and N57 for Nb2 (both in Chain C and D), and R52 (Chain D only). The binding interface may be rigid, accounting for a good quality of the electron density.

Minor points:

- The color used for depicting the published apo-opsin structure (PDB ID: 5TE3) in Fig. 4d is dark blue (also according to the fig caption). However, the authors state that the color is green in the following sentence: "By contrast, the comparison of the opsin/Nb2 structure (Fig. 4d, cyan) to a published apo-opsin crystal structure (Fig. 4d, green) showed..."

- Moreover, in Fig. 4e, it appears that the intracellular end of TM6 of the published apo-opsin complex is highlighted in sky blue or cyan instead of dark blue.

- In the Abstract sentence "...prevented outward movement of helices five and six – a universal activation event for GPCRs", the authors could specify that this outward movement occurs at the intracellular receptor side, as the movements at the extracellular side can be different or opposite.

Reviewer #5 (Remarks to the Author)

I reviewed the X-ray crystallographic work only. The authors report crystal structures of the complexes between Nb2 and bRho, bRho*, and opsin. These complexes reveal the binding interface between Nb2 and the receptor, and suggest the structural basis for the allosteric modulation of the receptor by Nb2. Even though the resolution of the structures is not very high (3.70, 4.25, and 3.71 Å), they are important for the present study and represent a substantial advancement in the field.

The X-ray crystallographic work appears solid. The chosen resolution cut-offs are supported by the CC1/2 values > 20%, 100% completeness, and good data redundancy (~10) in the highest resolution shell. Only the authors can assess whether the inclusion of reflections to the reported resolution limits is beneficial for the quality of the electron density despite other indicators (e.g., I / σ) being somewhat outside of the optimal range.

More than statistical indicators, the quality of the electron density matters. The authors discuss interactions between some of the GPCR and Nb2 residues in detail for the 3.70-Å complex. Given the not-so-high resolution, I suggest that the authors show in the Supplement the electron density (2Fo-Fc map, typically contoured at 1.0 σ) for the GPCR and Nb2 regions discussed in detail. I think that showing the quality of the electron density map is currently common practice in GPCR structure papers. It is nonetheless reassuring that according to the PDB Validation Report of the 3.70-Å complex, most of the discussed residues have a good fit to the electron density, except D56 and N57 for Nb2 (both in Chain C and D), and R52 (Chain D only). The binding interface may be rigid, accounting for a good quality of the electron density.

We appreciate the reviewer's summary and positive comments. We added a new Supplementary Fig. 2 showing the electron density (2Fo-Fc map, contoured at 1.0 σ) for the complexes in the three crystal structures and details of the regions of contact between (rhod)opsin and Nb2. Supplementary Figures 2-8 have been renumbered accordingly. We also clarified how side chain modeling was performed in the Methods section as follows:

“Side chains were modeled taking into consideration the density map and geometric plausibility, as well as their position in a higher resolution rhodopsin structure (PDB accession code: 1U19 [https://doi.org/10.2210/pdb4BEL/pdb], chain A)”

Minor

points:

- The color used for depicting the published apo-opsin structure (PDB ID: 5TE3) in Fig. 4d is dark blue (also according to the fig caption). However, the authors state that the color is green in the following sentence: “By contrast, the comparison of the opsin/Nb2 structure (Fig. 4d, cyan) to a published apo-opsin crystal structure (Fig. 4d, green) showed...”

We have corrected the error in the text.

- Moreover, in Fig. 4e, it appears that the intracellular end of TM6 of the published apo-opsin complex is highlighted in sky blue or cyan instead of dark blue.

TM6 in apo-opsin is dark blue, but the reflection effect of the flat end of the helix gives an appearance of light blue color. This artifact has been solved by removing the “specular reflection” effect in the updated figure.

- In the Abstract sentence "...prevented outward movement of helices five and six – a universal activation event for GPCRs", the authors could specify that this outward movement occurs at the intracellular receptor side, as the movements at the extracellular side can be different or opposite.

Thank you for the suggestion. It has been implemented in the revised abstract.